# Fixed Budget is No Harder Than Fixed Confidence in Best-Arm Identification up to Logarithmic Factors

**Kapilan Balagopalan** [* 1]   **Yinan Li** [* 1]   **Yao Zhao** [1]   **Tuan Ngo Nguyen** [1]   **Anton Daitche** [2]   **Houssam Nassif** [2]
**Kwang-Sung Jun** [3]

## Abstract

The best-arm identification (BAI) problem is one of the most fundamental problems in interactive machine learning, which has two flavors: the fixed-budget setting (FB) and the fixed-confidence setting (FC). For $K$-armed bandits with a unique best arm, the optimal sample complexities for both settings have been settled down and they match up to logarithmic factors. This prompts an interesting research question about the generic, potentially structured BAI problems: is FB harder than FC or the other way around? In this paper, we show that FB is no harder than FC up to logarithmic factors. We do this constructively: we propose a novel algorithm called FC2FB (fixed confidence to fixed budget), which is a meta algorithm that takes in an FC algorithm $\mathcal{A}$ and turn it into an FB algorithm. We prove that FC2FB enjoys a sample complexity that matches, up to logarithmic factors, that of the sample complexity of $\mathcal{A}$. This means that the optimal FC sample complexity is an upper bound of the optimal FB sample complexity up to logarithmic factors. Our result not only reveals a fundamental relationship between FB and FC, but also has a significant implication: FC2FB combined with existing state-of-the-art FC algorithms, leads to improved sample complexity for a number of FB problems.

## 1. Introduction

The best-arm identification (BAI) problem, also known as pure exploration in multi-armed bandits, is one of the most basic forms of interactive machine learning. BAI has been successful in many applications including A/B testing (Fiez et al., 2024), multivariate testing (Katz-Samuels et al., 2020),

heteroskedastic variance designs (Weltz et al., 2023), and hyperparameter optimization (Li et al., 2018). In BAI, the learner is given $K$ arms. Each arm has an unknown reward distribution – in particular, the mean reward is unknown and can be different across the arms. The learner sequentially interacts with the arms. At each time step $t$, the learner chooses an arm and receives a reward drawn from the chosen arm's reward distribution. The goal is to identify the arm with the highest mean reward as accurately as possible. Note that BAI can also be equipped with structural assumptions (Huang et al., 2017). For example, linear bandits assume that the mean reward of each arm is a linear function of its observed feature vector.

BAI comes with two flavors: the fixed-confidence setting (FC) (Even-Dar et al., 2006), and the fixed-budget setting (FB) (Audibert et al., 2010). In FC, the learner is given the target failure rate $\delta \in (0, 1)$ and pulls arms as many times as she wants until she can output an arm $\hat{J} \in [K]$ with a correctness guarantee: $\mathbb{P}(\hat{J} \neq 1) \leq \delta$ (where 1 is the index of the unique best arm). The performance of an FC algorithm is typically analyzed by how fast it stops. In FB, the learner is given a sampling budget $B$ instead of the target failure rate and pulls arms at most $B$ times in total. Then, she outputs an estimated best arm $\hat{J}$. Unlike FC, the FB setting does not require the learner to provide a certification. While FC and FB have different protocols and performance criteria, the sample efficiency of the algorithm can both be summarized as so-called sample complexity. Informally, the sample complexity is the amount of samples (i.e., budget) required until the estimated best arm $\hat{J}$ is correct with probability at least $1 - \delta$ where $\delta$ is input to the algorithm for FC but can be set arbitrarily for FB. In this way, one can compare the sample efficiency guarantees from FC and FB algorithms.

This raises an interesting research question:

*Does there exist a relationship between FC and FB? That is, is one problem fundamentally easier/harder than the other?*

There are mixed observations in the literature. In the standard $K$-armed bandit setting (unstructured), the optimal sample complexity of FC (Garivier & Kaufmann, 2016) and

[1] University of Arizona [2] Meta Inc [3] POSTECH CSE/GSAI. Correspondence to: Kapilan Balagopalan <kapilanbgp@arizona.edu>, Yinan Li <yinanli@arizona.edu>.

*Proceedings of the 43rd International Conference on Machine Learning*, Seoul, South Korea. PMLR 306, 2026. Copyright 2026 by the author(s).

FB (Katz-Samuels & Jamieson, 2020; Zhao et al., 2023) do not exactly match, and there is a logarithmic-factor gap. Turns out, the gap is not just an artifact of analysis. Carpentier & Locatelli (2016) report that there is a set of problem instances where FB algorithms must suffer an extra logarithmic factor in the sample complexity compared to FC algorithms. For generic structured problems, it is unclear if the same pattern will persist or if such a logarithmic factor difference can be exacerbated to a polynomial factor. Furthermore, there are many BAI problems where the best-known FB sample complexity is orderwise smaller than the FC counterpart; e.g., $K$-armed bandits with heterogeneous noise, linear bandits, etc (see Section 5 for details). On the other hand, in principle, compared to FB, FC has an extra verification/certification requirement (i.e., having to provide a correctness guarantee) that is typically known to be a barrier to having a further improved sample complexity in various situations (Katz-Samuels et al., 2020)

In this paper, we provide an answer to the research question raised above: the sample complexity of FB is no larger than that of FC, up to logarithmic factors. We achieve this by proposing a novel reduction algorithm called FC2FB (Fixed Confidence to Fixed Budget) that takes in an FC algorithm with a sample complexity guarantee of $T_\delta^* = A \ln(1/\delta)$ for some $A$ and turns it into an FB algorithm with a sample complexity of

$$A \ln(A/\delta) \cdot \ln\ln(1/\delta) .$$

FC2FB does so without requiring knowledge of $A$. This has a significant implication: an FC sample complexity can be transferred to an FB sample complexity up to polylog$(A, \ln(1/\delta))$, or simply, up to polylog$(T_\delta^*)$. Thus, *the optimal FC sample complexity is an upper bound of the optimal FB sample complexity up to logarithmic factors, leading to the conclusion that FB is no harder than FC up to logarithmic factors.* We describe FC2FB and its guarantee in Section 3. We also show that we can even weaken the requirement on the FC algorithm to having a guarantee for a particular constant $\delta$ rather than any $\delta$ in Section 4.

FC2FB not only reveals the fundamental relationship between FB and FC, but also provides a useful algorithmic construction in FB. We showcase that FC2FB can be applied to numerous FB BAI problems to improve existing sample complexity guarantees, including heterogeneous noise bandits, linear bandits, unimodal bandits and cascading bandits (Section 5). Finally, we also show our empirical study where our procedure can indeed outperform existing FB algorithms in Section 6. Important related works are cited and discussed throughout the paper, and other related works are discussed in Appendix A. We conclude with exciting future research directions in Section 7.

Our result should be interpreted as a non-asymptotic reduction between sample-complexity guarantees. It does not

---

**Algorithm 1** The fixed budget setting (FB)

1: **Input:** Sampling budget $B$
2: **for** $t = 1, 2, \ldots, B$ **do**
3:    The learner chooses an arm $I_t \in [K]$.
4:    The environment generates a reward $R_t \sim \nu_{I_t}$
5:    The learner observes $R_t$.
6: **end for**
7: The learner chooses an estimated best arm $\hat{J} \in [K]$
8: **Output**: $\hat{J}$

---

assert the existence of a universally exponent-optimal fixed-budget algorithm, a question known to be subtle in light of recent minimax (Komiyama et al., 2022) and no-complexity results (Degenne, 2023) for fixed-budget BAI.

## 2. Preliminaries

**Notations.** We denote $a_1, \ldots, a_n$ by $a_{1:n}$. We define $a \vee b := \max\{a, b\}$ and $a \wedge b := \min\{a, b\}$. We add a tilde to the standard big-O notations to denote that we are ignoring logarithmic factors; e.g., $\tilde{O}(\cdot), \tilde{\Theta}(\cdot), \tilde{\Omega}(\cdot)$, etc.

**The structured best-arm identification problem.** We consider the best-arm identification (BAI) problem in structured bandits. We are given $K$ arms. Each arm $i \in [K]$ is associated with an unknown reward distribution $\nu_i$ over $\mathbb{R}$. Let $X_{i,t} \sim \nu_i$ denote the reward obtained from the $t$-th pull of arm $i$; rewards are assumed i.i.d. across pulls of the same arm. For each arm $i$, we write $\mu_i := \mathbb{E}_{X \sim \nu_i}[X]$ for its (unknown) mean. Without loss of generality, we assume that $\mu_1 > \mu_2 \geq \cdots \geq \mu_K$, so arm 1 is the unique optimal arm. We define the suboptimality gap $\Delta_i := \mu_1 - \mu_i$.

In structured bandits, we assume that the bandit instance $\nu := (\nu_1, \ldots, \nu_K)$ belongs to a known model class $\mathcal{M}_K$ of admissible instances. This model class can encode arbitrary distributional properties, including constraints on means, variances, tails, supports, or structural relationships across arms, and is known to the learner. For example, the unstructured case corresponds to $\mathcal{M}_K$ being the set of all $K$-tuples of distributions in some base family (e.g., all distributions supported on $[0, 1]$ or being 1-sub-Gaussian). A linear bandit model can be expressed by taking $\mathcal{M}_K$ to be the set of instances for which there exists $\theta^* \in \mathbb{R}^d$ such that $\mu_i = \langle x_i, \theta^* \rangle$ for all $i \in [K]$, where the feature vectors $x_i \in \mathbb{R}^d$ are known to the learner. More generally, $\mathcal{M}_K$ may capture other structured families beyond linearity.

**The fixed-budget setting (FB).** In FB (Algorithm 1), the learner is given a total arm pull budget of $B$. For each time step $t$, the learner chooses an arm, pulls it, and then receives a reward sampled from its reward distribution. At the end of $B$-th time step, the learner must output its estimated best arm $\hat{J}$. We summarize the protocol in Algorithm 1.

We consider the standard error probability proposed in Au-

dibert et al. (2010):

$$\mathbb{P}(\hat{J} \neq 1) \, ,$$

i.e., the probability of best arm mis-identification, which is the smaller the better. For example, Successive Rejects algorithm (Audibert et al., 2010) results in a bound of $K^2 \exp(-\frac{B}{H})$ for some instance-dependent complexity measure $H$. For the sake of comparison, we found it easier to convert the error probability guarantee into the sample complexity.

**Proposition 2.1.** *Suppose that an FB algorithm satisfies* $B \geq B_0 \implies \mathbb{P}(\hat{J} \neq 1) \leq F \exp(-\frac{B}{H})$. *Then, there exists* $B' = \Theta(H \ln(F/\delta) + B_0)$ *such that if* $B \geq B'$ *then* $\mathbb{P}(\hat{J} \neq 1) \leq \delta$.

*Proof.* Set $F \exp(-\frac{B}{H}) = \delta$ and solve it for $B$, along with taking a maximum with $B_0$ since otherwise the guarantee is not true. $\square$

The value of $B'$ above is what we call the sample complexity in FB. Note that $B_0$ takes the role of "warm up" in that it does not scale with $\ln(1/\delta)$.

**The fixed-confidence setting (FC).** In FC (Algorithm 2), the learner is given a target failure rate $\delta \in (0, 1)$. We describe the protocol in Algorithm 2. The sampling steps

---

**Algorithm 2** The fixed-confidence setting (FC)

1: **Input:** Failure rate $\delta$
2: **for** $t = 1, 2, \ldots$ **do**
3:     The learner chooses an arm $I_t \in [K]$.
4:     The environment generates a reward $R_t \sim \nu_{I_t}$
5:     The learner observes $R_t$.
6:     **if** the stopping condition of the learner is true **then**
7:         $\tau \leftarrow t$     // stopping time
8:         The learner chooses an estimated best arm $\hat{J} \in [K]$.
9:         **break**
10:    **end if**
11: **end for**
12: **Output**: $\hat{J}$

---

are equivalent to FB, except that there is no hard limit on the number of samples the learner can take. Instead, the learner decides to terminate itself if she can verify which arm is the best with probability at least $1 - \delta$. Unlike FB, FC has a strict requirement on the algorithm: the correctness. We say that an FC algorithm is $\delta$-correct if it satisfies

$$\mathbb{P}(\hat{J} \neq 1, \tau < \infty) \leq \delta \, . \tag{1}$$

Since samples are costly, the main interest in FC is the stopping time $\tau$ defined in Algorithm 2, which is a random variable. We choose the high probability sample complexity as the performance metric of FC. In particular, we say that

an FC algorithm has a high probability sample complexity of $T_\delta^*$ if:

$$\mathbb{P}(\tau > T_\delta^*) \leq \delta. \tag{2}$$

In this paper, $\delta$ is the same value that is given as input to the learner, though it does not have to in general.

Establishing high-probability sample complexity bounds is a standard and widely adopted practice in the fixed-confidence literature. Just to name a few, Jamieson et al. (2014); Karnin et al. (2013); Even-Dar et al. (2006); Kaufmann & Kalyanakrishnan (2013); Katz-Samuels et al. (2020); Zhong et al. (2020).

## 3. From Fixed Confidence to Fixed Budget

We define two variants of fixed-confidence algorithms: strong and weak (A detailed discussion of these variants and their relation to existing methods with different types of guarantees is provided in Appendix E. A curious reader may consult it, but it can be skipped without loss of continuity for the following material). We focus exclusively on strong fixed-confidence algorithms in this section and defer the definition of weak fixed-confidence algorithms to Section 4. Basically, the strong fixed-confidence best arm identification algorithm is capable of satisfying the high probability sample complexity of order $\ln(1/\delta)$ for any $\delta$. In this section, we describe our proposed algorithm called FC2FB (Fixed Confidence to Fixed Budget) and provide theoretical guarantees.

FC2FB is a meta algorithm that takes in an FC algorithm and turn it into an FB algorithm, allowing us to transfer the sample complexity guarantee from FC to FB. Due to being a meta algorithm, we need to define what guarantee we expect from the given FC algorithm. For simplicity, we will first focus on the following definition of strong FC algorithms.

**Definition 3.1** (Strong FC algorithm). An FC algorithm is referred to as a strong FC algorithm if, for any $\delta \in (0, 1)$, it satisfies $\delta$-correctness (see (1)) and has a high probability sample complexity of $T_\delta^* = A \ln(1/\delta) + C$ (see (2)) for some $A$ and $C$ independent of $\delta$.

Essentially, strong FC algorithms are those that have a logarithmic dependency on $1/\delta$ rather than, say, a polynomial dependency. We will later show that it is possible to relax this condition so we can allow a polynomial dependence on $1/\delta$, or even to having the guarantees for a numerical constant $\delta$ only.

**Warm up: A naive framework for known $A$ and $C$:** As a warmup, we present a simple framework that takes a Strong FC algorithm with known $A$ and $C$ as its input and delivers an FB algorithm with a sample complexity no larger than FC algorithm *including logarithmic factors*.

To see this, first notice that for any budget $B$, we can find the $\delta_B$, such that $B = T^*_{\delta_B} = A \log \frac{1}{\delta_B} + C$.

Now we design an FB algorithm such that, for a given budget $B$, we run FC($\delta_B$). When the budget runs out we test if the FC algorithm has been self-terminated. If yes, we return the arm recommended by FC denoted by $J_{FC}$. Otherwise, we return an arbitrary arm.

Then,

$$\mathbb{P}(J_{FB} \neq 1) = \mathbb{P}(\text{FC stops and } J_{FC} \neq 1)$$
$$+ \mathbb{P}(\text{FC does not stop and } J_{FC} \neq 1)$$
$$\leq \delta_B + \delta_B = 2\delta_B.$$

To obtain the sample complexity for this FB algorithm, we need to compute how large the budget $B$ has to be to control the bound above to be at most $\delta$ (i.e., $\mathbb{P}(J_{FB} \neq 1) \leq 2\delta_B \leq \delta$). The computed sample complexity is:

$$A \log \frac{2}{\delta} + C = A \log \frac{1}{\delta} + C + A \ln 2.$$

Hence, it is clear that for known $A, C$, there exists an FB algorithm with the same sample complexity up to constant factors, and the term with $\ln(1/\delta)$ has the matching constant. However, our procedure is impractical because $A$ and $C$ are problem dependent constants unknown to the learner.

**The FC2FB algorithm.** We now construct a meta algorithm that has the same effect as the naive framework above except that we now do not require the knowledge of the problem-dependent constants. We present FC2FB in Algorithm 3. FC2FB takes in a strong FC algorithm $\mathcal{A}$ as input and executes it repeatedly over $R$ stages, each with roughly $B/R$ allocated samples. The main idea is that we try a wide range of $\delta$'s while force-terminating runs that do not self-terminate before using up the allocated samples.

Specifically, we increase failure rates $\delta$ at a doubly exponential rate over the stages. Since the stopping time of $\mathcal{A}$ scales with the input $\delta$, initial runs are likely to not finish within the desired budget. In this case, we force-terminate $\mathcal{A}$ and, clearly, we do not obtain any recommended arm. When we encounter the first stage in which $\mathcal{A}$ self-terminates and outputs a recommended arm $\hat{J}$, we stop the loop and finally output the recommended arm.

The parameters $R$ and $B'$ are chosen to achieve an exponentially decaying error rate w.r.t. the budget $B$ without requiring knowledge of $A$ and $C$ defined in Definition 3.1. Note that the hyperparameters $\delta_0$ can be set to be a constant like $1/2$ or $1/e$, and $Q$ is recommended to be set to $2K \ln K$. This is because if $Q$ is too small, then per-stage budget $B'$ would not be large enough to provide meaningful guarantees, though the guarantees will be true as long as $Q \leq B/2$.

This simple yet elegant algorithm achieves the following guarantee:

---

**Algorithm 3** FC2FB

1: **Input:** Total budget $B$, algorithm $\mathcal{A}$, base failure rate $\delta_0$, $Q \leq \frac{B}{2}$
2: Set $R := \lfloor \log_2(\frac{B}{Q}) \rfloor$, $B' := \lfloor \frac{B}{R} \rfloor$
3: $\hat{J} \leftarrow$ any arbitrary arm
4: **for** each stage $r = 1, 2, \ldots, R$ **do**
5: $\quad L_r = 2^{R-r}$
6: $\quad$ Run the algorithm $\mathcal{A}(\delta_0^{L_r})$ with the budget limit of $B'$
7: $\quad$ If the algorithm did not self-terminate, set $\hat{J}_r \leftarrow 0$; otherwise, let $\hat{J}_r$ be the output arm from the algorithm.
8: $\quad$ **if** $\hat{J}_r \neq 0$ **then**
9: $\quad\quad$ $\hat{J} \leftarrow \hat{J}_r$
10: $\quad\quad$ **break**
11: $\quad$ **end if**
12: **end for**
13: **Output:** $\hat{J}$

---

**Theorem 3.2** (Correctness of FC2FB). *For a strong fixed-confidence algorithm $\mathcal{A}$, given a total budget $B \geq$*
$$2\left( A \ln\left(\frac{1}{\delta_0}\right) + (C+1) \right) \ln\left( 2\frac{A}{Q} \ln\left(\frac{1}{\delta_0}\right) + \frac{2(C+1)}{Q} \right),$$
*$\delta_0 \leq 0.5$ and $Q \leq \frac{B}{2}$, algorithm FC2FB satisfies,*

$$\mathbb{P}\left( \hat{J} \neq 1 \right) \leq 3 \exp\left( -\frac{B}{\frac{4Q}{\ln(\frac{1}{\delta_0})} + 4A \log_2\left(\frac{B}{Q}\right)} \right).$$

Using an argument similar to Proposition 2.1, one can show that the theorem above with the choice of $\delta_0 = 1/e$ implies the sample complexity of

$$O\left( Q \ln(1/\delta) + A \ln(1/\delta) \cdot \ln(\frac{A \ln(1/\delta)}{Q}) + C \right).$$

While setting $Q$ to be $A$ would cancel out the extra factor of $\ln(A)$, one rarely has knowledge of $A$. Thus, our generic recommendation is to simply set $Q = 1$. Then, we obtain a sample complexity bound of order $A \ln(1/\delta) + C$ up to polylog$(A, \ln(1/\delta))$ factors or simply, up to polylog$(T^*_\delta)$.

In practice, we recommend that $Q$ be set to the minimal number of samples for the algorithm $\mathcal{A}$ to enjoy any meaningful guarantee. For example, for the unstructured $K$-armed bandit problem, set $Q = K$, and for linear bandits, set $Q = d$.

It is noteworthy that in the above guarantee, the factor $A$ from the high-probability sample complexity bound of the strong fixed-confidence algorithm is preserved in the correctness guarantee of the fixed-budget setting, up to logarithmic factors. Hence, this opens the door to numerous applications where there exists a gap between the sample complexity bounds of fixed-confidence and fixed-budget settings. We can use our algorithm to bridge this gap up to logarithmic factors. We discuss these applications in Section 5. The

proof of Theorem 3.2 is deferred to Appendix B.

**Fixed-confidence to anytime fixed-budget:** Furthermore, we present another algorithm, FC2AT, that transforms a strong fixed-confidence algorithm into an anytime algorithm. In the anytime setting, for any fixed deterministic time $t$ that is unknown for the learner, the agent aims at achieving a low probability of error at time $t$ (Jun & Nowak (2016), Zhao et al. (2023)). While $B$ is fixed and known in the fixed-budget setting, $t$ is fixed and unknown in the anytime setting. The analysis and guarantees of FC2AT are deferred to Appendix D.

## 4. From Weak to Strong Fixed-Confidence Algorithms

In the previous section, we conditioned that the FC algorithm required by FC2FB has to be a strong FC. In this section, we establish that this condition can be relaxed. In fact, we can transform a weak fixed-confidence algorithm into a strong fixed-confidence algorithm using FCW2S, a meta framework inspired by the Brakebooster algorithm of Balagopalan et al. (2025). Before proceeding, we define the weak fixed-confidence algorithm. Essentially, a weak fixed-confidence algorithm either satisfies correctness and high-probability sample complexity guarantees only for **some** error rate $\delta$, or the high probability sample complexity lacks dependence on $\ln(1/\delta)$.

**Definition 4.1** (Weak Fixed-Confidence Algorithm). An FC algorithm $\mathcal{A}^w$ is referred to be a weak FC algorithm if, for **some** $\delta_0 \in (0,1)$, it satisfies $\delta_0$-correctness (see (1)) and has a high probability sample complexity of $T_{\delta_0}^* = f(\delta_0)$ (see (2)), where $f(\delta_0)$ denotes some function of $\delta_0$ that does not necessarily have logarithmic dependence on $1/\delta_0$.

Now, we present a framework to transform any weak fixed-confidence algorithm into a strong fixed-confidence algorithm. The framework, *FCW2S* (Fixed-Confidence Weak to Strong), is presented in Algorithm 4. FCW2S runs $L$ independent instances of a weak FC algorithm ($\mathcal{A}^w$) with $\delta_0$ failure rate in parallel, i.e, each instance gets to pull the arm in a round-robin style allocation. If any of these instances self-terminates (stopping condition of the $\mathcal{A}^w$ met), we record the output arm $\hat{J}_\ell$ (this is one vote for that arm) and eliminate the instance from the round-robin. If $\lfloor L/2 \rfloor$ instances self-terminate, then we stop the round-robin allocation and start to count votes. Here, the number of votes $v_i$ means the number of instances that output arm $i$. Whoever wins this voting becomes the final output $\hat{J}$.

One can relate this algorithm to the so-called *boosting* or *amplification*, where weak learners are used to create a strong guarantee. Here, we also use weak FC to create a strong FC. When we use multiple instances, the probability of getting wrong output from more than half of the instances

---

**Algorithm 4** FCW2S

**Input:** algorithm $\mathcal{A}^w$, the number of trials $L$, base failure rate $\delta_0$

**Initialize**: $L$ instances of $\mathcal{A}^w$: $\mathcal{A}_1^w, \ldots, \mathcal{A}_L^w$

Set the surviving set: $\mathcal{S} \leftarrow \{\mathcal{A}_1^w, \ldots, \mathcal{A}_L^w\}$

**while** $|\mathcal{S}| \geq \lfloor L/2 \rfloor$ **do**

    Choose $\mathcal{A}_\ell^w \in \mathcal{S}$ with the smallest internal time step.

    Choose an arm $J$ recommended by $\mathcal{A}_\ell^w$ and receive a reward $R$.

    Send $(J, R)$ to $\mathcal{A}_\ell^w$ for its update.

    **if** $\mathcal{A}_\ell^w$ terminates **then**

        Set $\hat{J}_\ell \leftarrow$ the arm output from $\mathcal{A}_\ell^w$.

        $\mathcal{S} \leftarrow \mathcal{S} \backslash \{\mathcal{A}_\ell^w\}$

    **end if**

**end while**

Count the votes:

$$v_i = \sum_{\ell=1}^L \mathbb{1}\{\ell \notin \mathcal{S}, \hat{J}_\ell = i\}, \ \forall \ i \in [K]$$

**Output**: $\hat{J} \leftarrow \arg\max_{i \in [K]} v_i$

---

decays exponentially.

FCW2S achieves the following correctness and sample complexity guarantees.

**Proposition 4.2** (Correctness of $FCW2S(\delta)$). *For any error rate $\delta \in (0,1)$, for any weak fixed-confidence algorithm $\mathcal{A}^w$ that satisfies Definition 4.1 with $\delta_0 < \frac{1}{4e}$, running the fixed confidence algorithm FCW2S with $L \geq 4 \frac{\ln \frac{1}{\delta}}{\ln \frac{1}{4e\delta_0}}$ satisfies $\mathbb{P}\left(\hat{J} \neq 1, \tau < \infty\right) < \delta$.*

**Proposition 4.3** (Strong stopping time guarantee of $FCW2S(\delta)$). *For any error rate $\delta \in (0,1)$, for any weak fixed-confidence algorithm $\mathcal{A}^w$ that satisfies Definition 4.1 with $\delta_0 < \frac{1}{4e}$, running the fixed confidence algorithm FCW2S with $L \geq 4 \frac{\ln \frac{1}{\delta}}{\ln \frac{1}{4e\delta_0}}$ satisfies, ($\tau$ denotes the stopping time)*

$$\mathbb{P}\left(\tau > L \cdot f(\delta_0)\right) < \delta.$$

*Furthermore, set $L = 4 \frac{\ln \frac{1}{\delta}}{\ln \frac{1}{4e\delta_0}}$, then,*

$$\mathbb{P}\left(\tau > \left(4\frac{f(\delta_0)}{\ln \frac{1}{4e\delta_0}}\right) \cdot \ln \frac{1}{\delta}\right) < \delta.$$

Propositions 4.2 and 4.3 show that any weak fixed-confidence algorithm can be transformed into a strong fixed-confidence algorithm by carefully choosing $L$. This is particularly useful when an algorithm lacks a stopping time guarantee that depends on $\log(1/\delta)$ but satisfies Definition 4.1 (e.g., with polynomial dependence on $\delta$). In such cases, we can use that algorithm as an input to FCW2S and tune $L$ to achieve stronger guarantees. Proof of the Propositions 4.2 and 4.3 are deferred to Appendix C.

**Algorithm 5** Phased elimination for known heterogeneous noise (PE-KHN)

1: **Input:** Target failure rate $\delta$
2: $\mathcal{S}_1 \leftarrow [K], \ell \leftarrow 1$
3: **while** $|\mathcal{S}_\ell| > 1$ **do**
4:      $\varepsilon_\ell \leftarrow 1/2^\ell, \delta_\ell \leftarrow \frac{1}{\ell(\ell+1)} \cdot \delta$
5:      For every arm $i \in S_\ell$, collect additional samples so that the total sample size for arm $i$ is $\lceil \frac{2\sigma_i^2}{\varepsilon_\ell^2} \ln(\frac{K}{\delta_\ell}) \rceil$.
6:      For every arm $i \in S_\ell$, compute the sample mean $\hat{\mu}_i^{(\ell)}$.
7:      $\mathcal{S}_{\ell+1} \leftarrow \mathcal{S}_\ell \setminus \{i \in S_\ell \mid \hat{\mu}_i^{(\ell)} \leq \max_{j \in S_\ell} \hat{\mu}_j^{(\ell)} - 2\varepsilon_\ell\}$
8:      $\ell \leftarrow \ell + 1$
9: **end while**
10: Output: The only remaining arm in $\mathcal{S}_\ell$

By now, we have addressed the questions: *Does there exist a relationship between FC and FB? That is, is one problem easier/harder than the other?* YES, there exists a relationship between FC and FB and FB is no harder than FC (Section 3); *Is a strong fixed-confidence algorithm a necessary requirement for FC2FB?* NO, we can use weak FC as well (Section 4). In the next section, we discuss several applications of FC2FB instantiated with both strong and weak FC algorithms. That is being said, we would like to acknowledge that, as of now, although FC2FB can be instantiated algorithmically, it cannot be analyzed with FC algorithms that have only asymptotic stopping time guarantees (e.g., Track-and-Stop Garivier & Kaufmann (2016)).

## 5. Applications

FC2FB not only reveals the fundamental relationship between FC and FB, but also provides a useful tool for constructing new FB algorithms with superior guarantees than existing ones. FC2FB achieves this by transferring sample complexity guarantees of superior FC algorithms to FB domains. In this section, we provide case studies demonstrating such examples. Hereafter, we denote by FC2FB($\mathcal{A}$) the instantiation of FC2FB with $\mathcal{A}$ as the input FC algorithm.

### 5.1. Known Heterogeneous Noise

In this section, we consider the case where each arm $i$ has a $\sigma_i^2$-sub-Gaussian reward distribution where each $\sigma_i$ is known to the learner. For FC, it is easy to adapt a standard elimination algorithm by adjusting the sample assignment for each arm based on its noise level, which we present in Algorithm 5. Algorithm 5 attains the following sample complexity guarantee.

**Theorem 5.1** (The sample complexity of Algorithm 5). *Assume* $\Delta_j \leq 1 \quad \forall j \in \mathcal{S}$ *and define* $T_\delta^* := \frac{64\sigma_1^2}{\Delta_2^2} \ln \left( \frac{4K(\ln 2)^2}{\delta} \ln^2(4/\Delta_2) \right) +$

*Table 1.* Sample complexity comparison for various FB algorithms, for sub-Gaussian reward distribution with known noise level. $\tilde{O}(\cdot)$ ignores logarithmic factors.

| Algorithm | Sample complexity |
|---|---|
| SHVar (Lalitha et al., 2023) | $\tilde{O}\left( \frac{\sigma_1^2}{\Delta_2^2} + \sum_{i=2}^K \frac{\sigma_i^2}{\Delta_2^2} \right)$ |
| SH (Karnin et al., 2013) | $\tilde{O}\left( \frac{\sigma_1^2}{\Delta_2^2} + \sum_{i=2}^K \frac{\sigma_1^2 + \sigma_i^2}{\Delta_i^2} \right)$ |
| FC2FB(PE-KHN) (ours) | $\tilde{O}\left( \frac{\sigma_1^2}{\Delta_2^2} + \sum_{i=2}^K \frac{\sigma_i^2}{\Delta_i^2} \right)$ |

$\sum_{j\neq 1}^K \frac{64\sigma_j^2}{\Delta_j^2} \ln \left( \frac{4K(\ln 2)^2}{\delta} \ln^2(4/\Delta_j) \right)$. *Then, Algorithm 5 satisfies*

$$\mathbb{P}(\tau > T_\delta^*) \leq \delta .$$

Proof of Theorem 5.1 is deferred to Appendix F. Ignoring the $\ln(\ln(\cdot))$ factors, we have $T_\delta^* = O\left( \left( \frac{\sigma_1^2}{\Delta_2^2} + \sum_{i=2}^K \frac{\sigma_i^2}{\Delta_i^2} \right) \cdot \ln \frac{K}{\delta} \right)$, which matches the lower bound established in Lu et al. (2021) up to $\ln K$ terms. Therefore, FC2FB can take Algorithm 5 as input and transform it into an FB algorithm whose sample complexity reflects the guarantee established in Theorem 5.1. This results in performance superior to state-of-the-art fixed-budget algorithms for known heterogeneous noise settings. See Table 1.

**Corollary 5.2** (Error probability bound for FC2FB(PE-KHN)). *Given a total budget* $B \geq 2A\ln(4A)$ *where,* $A = \Theta\left( \left( \frac{\sigma_1^2}{\Delta_2^2} + \sum_{i=2}^K \frac{\sigma_i^2}{\Delta_i^2} \right) \cdot \ln K \right)$, *by choosing* $\delta_0 = \frac{1}{e}$ *and* $Q = 1$, *algorithm FC2FB(PE-KHN) satisfies*

$$\mathbb{P}\left( \hat{J} \neq 1 \right) \leq 3\exp\left( -\frac{B}{4 + 4A\ln B} \right)$$

We show that our sample complexity from FC2FB(PE-KHN) can be order-wise smaller than that of SHVar by considering the following instance. Let $\Delta_2 = \frac{1}{K^5}$, $\Delta_i = \frac{1}{K^4}$ for $i = 3, \ldots, K$, $\sigma_1^2 = \sigma_2^2 = \frac{1}{K^5}$, $\sigma_i^2 = \frac{1}{K^2}$ for $i = 3, \ldots, K$. For this instance, our sample complexity from FC2FB is:

$$O\left( \frac{\sigma_1^2}{\Delta_2^2} + \sum_{i=2}^K \frac{\sigma_i^2}{\Delta_i^2} \right) = O\left( K^7 \right) ,$$

whereas the sample complexity from SHVar yields:

$$O\left( \frac{\sigma_1^2}{\Delta_2^2} + \sum_{i=2}^K \frac{\sigma_i^2}{\Delta_2^2} \right) = O\left( K^9 \right) .$$

### 5.2. Heterogeneous Noise with Unknown Variance

In this section, we consider the case where the reward distributions of the arms are bounded with unknown variances. Fixed-confidence algorithm VD-BESTARMID

*Table 2.* Sample complexity comparison for various FB algorithms, for bounded reward distribution with unknown variances. $\tilde{O}(\cdot)$ ignores logarithmic factors.

| Algorithm | Sample complexity |
|---|---|
| VBR (Faella et al., 2020) | $\tilde{O}\left(\max\limits_{i=2\cdots K} \frac{\sigma_1^2+\sigma_i^2}{\Delta_i^2} \cdot K\right)$ |
| SHAdaVar (?) | $\tilde{O}\left(\frac{\sigma_{max}^2}{\Delta_2^2} \cdot K\right)$ |
| FC2FB(VD-BESTARMID) (ours) | $\tilde{O}\left(\sum_{i=1}^{K}\left(\frac{\sigma_i^2}{\Delta_i^2} + \frac{1}{\Delta_i}\right)\right)$ |

from Lu et al. (2021) achieves samples complexity of $O\left(\sum_{i=1}^{K}\left(\frac{\sigma_i^2}{\Delta_i^2} + \frac{1}{\Delta_i}\right)\ln\delta^{-1}\right)$ with $\Delta_1 := \Delta_2$ for this case, ignoring doubly logarithmic factors. We apply FC2FB to VD-BESTARMID and achieve the following guarantee,

**Corollary 5.3** (Error probability bound for FC2FB(VD-BESTARMID))**.** *For algorithm VD-BESTARMID, given a total budget $B \geq 2A\ln(4A)$ where,* $A = O\left(\sum_{i=1}^{K}\left(\frac{\sigma_i^2}{\Delta_i^2} + \frac{1}{\Delta_i}\right)\ln\delta^{-1}\right)$ *with $\Delta_1 := \Delta_2$, by choosing $\delta_0 = \frac{1}{e}$ and $Q = 1$, algorithm FC2FB satisfies*

$$\mathbb{P}\left(\hat{J} \neq 1\right) \leq 3\exp\left(-\frac{B}{4+4A\ln B}\right)$$

In Table 2, we see that FC2FB(VD-BESTARMID) is better than SHAdaVar (?) by a factor of $K$, when $\Delta_2 \ll \Delta_3 \leq \cdots \leq \Delta_K$ and better than VBR (Faella et al., 2020) by a factor of $K$, when $\sigma_1 \gg \sigma_i$, $\forall K \geq i > 1$.

The improvement comes from transferring an FC guarantee whose leading term depends on the individual gap and variance of each arm, thereby improving upon the more conservative dependence of existing FB guarantees on global worst-case quantities, such as the largest variance or the smallest gap across all arms.

### 5.3. Linear Bandits

In the linear bandit problem, each arm $i$ is equipped with a feature vector $x_i \in \mathbb{R}^d$ and its reward distribution $\nu_i$ has the mean reward of $\langle x_i, \theta^* \rangle$ for some unknown $\theta^*$.

Katz-Samuels et al. (2020) report the best known sample complexity for FC linear bandits, which is

$$\gamma^* + \rho^* \ln(1/\delta) + d \quad (3)$$

where we ignore logarithmic factors except for $\ln(1/\delta)$ and both $\gamma^*$ and $\rho^*$ are some instance-dependent constants that are precisely defined in Katz-Samuels et al. (2020).

On the other hand, Katz-Samuels et al. (2020) propose an FB linear bandit algorithm whose sample complexity is

$$(\gamma^* + \rho^*)\ln(1/\delta) + d$$

where we again ignore logarithmic factors except for

$\ln(1/\delta)$. Note that $\gamma^*$ now comes with a logarithmic factor.

Using FC2FB, we improve the error probability bound in FB linear bandits as follows.

**Corollary 5.4.** *Consider FC2FB combined with the algorithm Fixed Confidence Peace (Katz-Samuels et al., 2020, Algorithm 1) with $\delta_0 = 1/e$ and $Q = 1$. This algorithm satisfies that, for some $B_0 = \tilde{\Theta}(\gamma^* + \rho^* + d)$,*

$$B \geq B_0 \implies \mathbb{P}(\hat{J} \neq 1) \leq c\exp\left(-\frac{B}{\tilde{\Theta}(\rho^*)\ln(B)}\right)$$

*for some numerical constant c.*

The bound above results in a matching sample complexity with (3), ignoring doubly logarithmic factors, which is a significant improvement.

As another comparison, Yang & Tan (2022) has shown that their FB algorithm called OD-LinBAI achieves the sample complexity of

$$H_{2,\text{lin}} \ln(1/\delta)$$

ignoring logarithmic factors except for $\ln(1/\delta)$, where $H_{2,\text{lin}} := \max_{2 \leq i \leq d} i\Delta_i^{-2}$. In the Appendix G, we show that $\rho^* = \tilde{O}(H_{2,\text{lin}})$. Together with the fact that $\gamma^* \leq c\rho^*\ln(K)$ for some numerical constant $c$ (Katz-Samuels et al. (2020), Proposition 1), our sample complexity is no larger than that of OD-LinBAI.

One may speculate that $H_{2,\text{lin}}$ might be of the same order as $\rho^*$. We show in our appendix G that this is not the case by constructing a set of instances where the difference between $\rho^*$ and $H_{2,\text{lin}}$ can be arbitrarily large.

### 5.4. Unimodal Bandits

In unimodal bandits, the arms are arranged along an ordered structure, and the mean rewards form a unimodal function of the indices: they first increase up to a single peak and then decrease. This structural assumption enables more efficient exploration compared to general multi-armed bandits, but it also leads to different complexities depending on the exploration objective. In the fixed-budget setting, the existing best upper bound in the literature on the sample complexity of best-arm identification is of order $\Delta^{-2}$ (Ghosh et al., 2024), which is governed by the global smallest gap $\Delta := \min_{2 \leq i \leq K} |\mu_i - \mu_{i-1}|$. In contrast, in the fixed-confidence setting, the optimal sample complexity is determined by the local information-theoretic quantity $T^*(\mu)$ (Poiani et al., 2024), which for Gaussian arms is of order $\sum_{i \in \mathcal{N}(*)}(\mu_* - \mu_i)^{-2}$, depending only on the neighbors of the best arm. Beyond asymptotic optimality, Poiani et al. (2024) also provides a finite-time upper bound on the expected stopping time in their Theorem 3.7, $\mathbb{E}_\mu[\tau_\delta] \leq T_\mu(\delta) + 15K$, where $T_\mu(\delta)$ is precisely defined in their Theorem 3.7.

As we will show in Proposition 5.8, $T_\mu(\delta) + 15K$ can be significantly smaller than $\Delta^{-2}$, implying that existing fixed-budget algorithms may require substantially more samples than fixed-confidence algorithms. This gap highlights the potential benefit of our FC2FB meta-algorithm: by leveraging the more refined guarantees available in the fixed-confidence setting, it offers a systematic way to obtain stronger fixed-budget algorithms for unimodal bandits. Along the way, we also showcase the usage of our FCW2S algorithm.

One can apply Markov's inequality to convert the expected sample complexity guarantee into a high-probability guarantee:

**Corollary 5.5** (Corollary of Theorem 3.7 in Poiani et al. (2024)). *Consider running UniTT in Section 3.4 in Poiani et al. (2024) with the input failure rate $\delta$ being a small constant and the other parameters specified in Theorem 3.7 in Poiani et al. (2024), UniTT is $\delta$-correct, and, with probability at least $1-\delta$, for all Gaussian instances with unit variance and unimodal means, the stopping time satisfies,*

$$\tau \le \frac{T_\mu(\delta) + 15K}{\delta},$$

*where $T_\mu(\delta)$ is precisely defined in their Theorem 3.7 in Poiani et al. (2024).*

Applying our weak-to-strong conversion algorithm FCW2S yields the following proposition:

**Proposition 5.6.** *For any error rate $\delta \in (0, 1)$, running FCW2S algorithm based on UniTT with a constant confidence level $\delta_1 < \frac{1}{4e}$, and number of trials $L \ge 4\frac{\ln \frac{1}{\delta}}{\ln \frac{1}{4e\delta_1}}$ (we call it FCW2S(UniTT, $\delta_1$, L)), we have:*

$$\mathbb{P}\left(\hat{J} \ne 1, \tau < \infty\right) < \delta.$$

*and*

$$\mathbb{P}\left(\tau > L \cdot \frac{T_\mu(\delta_1) + 15K}{\delta_1}\right) < \delta.$$

*Proof.* Combining Corollary 5.5 together with Propositions 4.2 and 4.3. $\square$

Finally, we are ready to present the result from applying FC2FB on FCW2S(UniTT, $\delta_1$, L).

**Corollary 5.7.** *Consider running Algorithm FC2FB with the base algorithm FCW2S(UniTT, $\delta_1 = \frac{1}{8e}$, $L = \lceil 4\frac{\ln \frac{1}{\delta}}{\ln \frac{1}{4e\delta_1}} \rceil$, base failure rate $\delta_0 = 1/e$, $Q = 1$ and the total budget $B$. This algorithm satisfies, for some $B_0 = \tilde{\Theta}(T_\mu(\delta_1) + K)$*

$$B \ge B_0 \implies \mathbb{P}(\hat{J} \ne 1) \le 3\exp\left(-\frac{B}{\tilde{\Theta}(T_\mu(\delta_1) + K)\ln B}\right)$$

We show in the next proposition that $\Delta^{-2}$ is of higher order than $T_\mu(\delta_1) + K$, up to logarithmic factors, which indicates

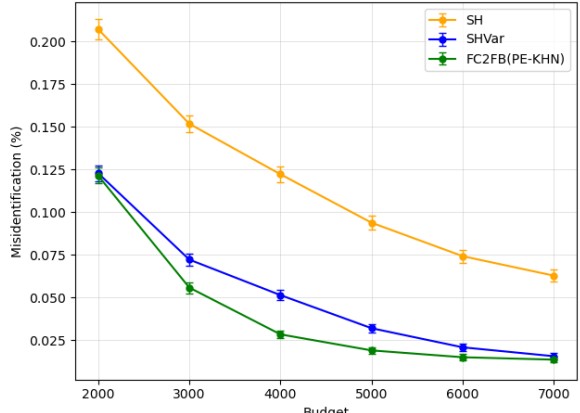

*Figure 1.* Probability of misidentifying the best arm in the Gaussian bandit as a function of the budget $B$, with $K = 32$ arms. Results are averaged over 5,000 trials.

our FC2FB framework can achieve a lower FB sample complexity compared to the existing best sample complexity in the literature for unimodal bandits. The proof of Proposition 5.8 is in Appendix H.

**Proposition 5.8.** *Suppose the unimodal means lies in $[0, 1]$, then*

$$T_\mu(\delta_1) + K = \tilde{O}(\Delta^{-2})$$

*and there exists an instance where the above two quantities are order-wise different, i.e., $T_\mu(\delta_1) + K = o(\Delta^{-2})$. The quantity $T_\mu(\delta)$ is defined in their Theorem 3.7 in Poiani et al. (2024) and $\Delta := \min_{2 \le i \le K} |\mu_i - \mu_{i-1}|$ as defined in Ghosh et al. (2024).*

## 6. Experiments

In this section, we empirically evaluate our proposed meta algorithm, FC2FB. We choose Sequential Halving (SH) (Karnin et al., 2013) and its known variance-adaptive version SHVar (Lalitha et al., 2023), discussed in the previous section. We show two experiment settings here and include a few other instances in the appendix J. Our experiment is on a Gaussian bandit with $K$ arms. The sub optimal gap of second arm is $\Delta_2 = 0.1$ and the sub optimal gap of other arm is $\Delta_i = 0.8$, $\forall i > 2$. We uniformly sample the variances in range of $[1, 2]$. In Figure 1, we report the probability of misidentifying the best arm among $K = 32$ arms as the budget increases. As expected, our proposed algorithm FC2FB(PE-KHN) outperforms both SH and SHVar except when the budget is 1000. This could possibly result from the $\ln(B)$ in the denominator of Theorem 3.2, which can be non-negligible at small budgets. For small budget regime $\ln(B)$ is comparable to $B$, however $B$ dominates in larger budget regime.

In our next experiment, we fix the budget $B$ at 6000 and vary the number of arms from 32 to 128. As shown in Figure 2, the probability of misidentifying the best arm

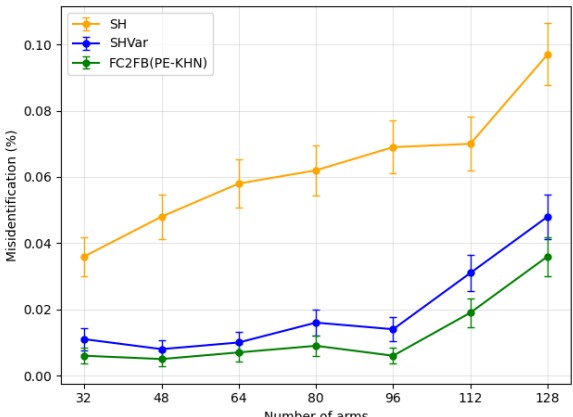

*Figure 2.* Probability of misidentifying the best arm in the Gaussian bandit as a function of the number of arms. The budget is fixed at 6000. Results are averaged over 5,000 trials.

increases as the number of arms $K$ grows. Our FC2FB(PE-KHN) demonstrates strong performance across a wide range of $K$, whereas SHVar performs well only when the number of arms is small and performs poorer as $K$ increases.

## 7. Conclusion

In this paper, we have shown that our proposed meta algorithm can take an FC algorithm and turn it into an FB algorithm with a comparable sample complexity. This reveals a fundamental relationship between FC and FB, indicating that FB is no harder than FC, up to logarithmic factors.

Our work opens up numerous interesting research problems. First, we have focused on the high probability guarantees only. It would be interesting to consider other performance measures such as simple regret and see if there are similar relationships. Second, our work assumes that there is a unique best arm. If such a condition is not met, it is desirable to focus on returning an $\varepsilon$-good arm. Since it is known that the FB setting can enjoy an $\varepsilon$-good arm guarantee for all $\varepsilon$ simultaneously with a single algorithm (Zhao et al., 2023), it would be interesting to see if there exists a meta algorithm that can turn a fixed-confidence algorithm with an $\varepsilon$-good arm guarantee into an FB algorithm that can have a guarantee for any $\varepsilon$. Finally, our reduction, by design, does not share the samples across different runs of the base FC algorithm, and thus tends to be less practical for small problem sizes. It would be interesting to develop more practical versions.

## Impact Statement

This paper presents work whose goal is to advance the field of Machine Learning. There are many potential societal consequences of our work, none of which we feel must be specifically highlighted here.

## Acknowledgments

Kapilan Balagopalan, Yinan Li, Yao Zhao, Tuan Ngo Nguyen, Kwang-Sung Jun were supported in part by the National Science Foundation under grant CCF-2327013 and Meta Platforms, Inc. This work was partly supported by Institute of Information & communications Technology Planning & Evaluation (IITP) grant funded by the Korea government (MSIT) (No.RS-2019-II191906, Artificial Intelligence Graduate School Program (POSTECH)).

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

# Appendix

## Table of Contents

# A. Related Work

The closest work in the literature that tries to connect FB and FC is Gabillon et al. (2012) that is restricted to the standard unstructured bandit problem. This work shows that there exists a unified algorithmic framework that uses the same arm selection and return strategies for FB and FC. However, their FB algorithm requires a very strong assumption that the learner has access to the problem-dependent complexity parameter $H_\varepsilon = \sum_{i \geq 2} \frac{1}{(\Delta_i \vee \varepsilon)^2}$. Furthermore, their work does not provide a blackbox reduction as we do – instead, they provide an algorithmic template and call it a meta algorithm, which is instantiated to FB or FC. Therefore, their work does not reveal any fundamental relationship between FB and FC that we do.

Another attempt appeared in Jun & Nowak (2016), which attempts to construct an (anytime) fixed-budget algorithm specifically based on LUCB, and whose analysis does not extend beyond LUCB. However, their approach suffers from a soundness issue (in particular their Lemma 7 and 8) that we believe is not possible to fix. Specifically, their algorithm waits until the LUCB algorithm satisfies the stopping condition and then restarts it with a smaller $\delta$, but the distribution of the stopping time (the time step satisfying the stopping condition) may not have a light enough tail to lead to an exponentially-decaying error probability. In fact, Theorem 2.5 in Balagopalan et al. (2025) showed that a certain version of LUCB's stopping time can be infinity with a constant probability. Our algorithm goes around this issue by employing a forced termination when the algorithm does not self-terminate within a predefined time horizon. Finally, our strategy is better than Jun & Nowak (2016) in that it works for any strong FC algorithm rather than being specific to LUCB. Furthermore, it works for the generic problem of BAI with structure (See Section 2), whereas Jun & Nowak (2016) only considers the unstructured version.

To compare UCB-E (Audibert et al., 2010) with ours, UCB-E requires the user to input a parameter $a$ whose efficient range depends on the complexity $H$ (sum of inverse squared gaps). In reality, $H$ is almost never available. Our FC2FB algorithm's input is just the budget $B$ and a base failure rate $\delta_0$ (e.g., can easily set to $1/e$ with no harm) – in particular, it does not require as input the problem complexity. While UCB-E has a tighter bound than ours if one knows $H$, our result is significant because it achieves comparable performance (up to log factors) without knowing $H$, solving the practical issue of unknown instance hardness.

Recent work has clarified why fixed-budget BAI should not be viewed as a direct analogue of fixed-confidence BAI. Degenne (2023) formalizes the notion of an asymptotic fixed-budget complexity, motivated by Qin (2022)'s COLT open problem, and studies whether a sufficiently large class of fixed-budget algorithms can admit an instance-wise complexity in the same spirit as fixed-confidence BAI. In fixed confidence, Track-and-Stop-type algorithms (e.g., Garivier & Kaufmann, 2016) can match an instance-dependent complexity asymptotically, and the optimal static-proportion oracle plays a central role. In fixed budget, however, Degenne (2023) shows that if an algorithm class containing all static-proportion algorithms admits such a complexity, then it must coincide with the oracle difficulty of static proportions; nevertheless, such a complexity does not exist for several basic tasks, including Gaussian BAI for sufficiently large $K$ and Bernoulli BAI already with two arms. Thus, unlike the fixed-confidence setting, no single fixed-budget algorithm can be uniformly exponent-optimal over all instances in these settings. Degenne (2023) also identifies remaining open cases, including thresholding bandits and Gaussian BAI for small $K > 2$.

Komiyama et al. (2022) study a complementary minimax formulation of fixed-budget BAI. They characterize minimax exponent rates through optimization problems, introduce the rates $R^{\text{go}}$ and $R^{\text{go}}_\infty$, and propose $R^{\text{go}}$-tracking as well as delayed optimal tracking (DOT). DOT asymptotically attains $R^{\text{go}}$ for Bernoulli and Gaussian arms, but computing or approximating the required optimization is difficult because it is over high-dimensional allocation functions; the paper explicitly leaves the existence of a computationally tractable, provably optimal algorithm as an important open problem.

Our work is orthogonal to these asymptotic exponent-optimality questions. We are not concerned with identifying the sharp fixed-budget error exponent or optimizing constant and logarithmic factors in the exponent. Rather, our results compare non-asymptotic fixed-confidence and fixed-budget sample complexities, up to logarithmic factors. Our result is also consistent with prior results on fixed-confidence and fixed-budget sample complexities in standard $K$-armed BAI. Carpentier & Locatelli (2016) show that fixed-budget algorithms can require an additional logarithmic factor in sample complexity compared with fixed-confidence algorithms on certain instances. The logarithmic overhead in FC2FB means that the sharp fixed-budget exponent problem, including the tractable attainment of the minimax rates studied by Komiyama et al. (2022) and the existence/nonexistence issues studied by Degenne (2023), remains open.

## B. A Meta-Algorithm for Fixed-Budget Conversion: FC2FB

In this section, we prove Theorem 3.2. Algorithm 3 is designed such that the budget $B$ is strategically split into each stage budget $B'$, with the strong fixed-confidence algorithm $\mathcal{A}$ executed at each stage using an exponentially increasing error rate. The intuition underlying this construction is as follows: if a stage with low $\delta$ self-terminates, the probability of error is correspondingly low. Conversely, if, only a stage with higher $\delta$ self-terminates, the correctness of the output cannot be guaranteed; however, given sufficient budget, it is improbable that all preceding stages with lower $\delta$ values failed to self-terminate.

**Definition B.1.** Let $B'$ denotes the per stage budget and $R$ denotes the number of stages in Algorithm 3, for some $\delta_0 < 1$, define

$$r^* = \min\{r \geq 1 : B' \geq T^*_{\delta_0^{L_r}}\}.$$

The threshold stage $r^*$ is defined as the point at which the allocated stage budget $B'$ first exceeds $T_{\delta_0^{L_{r^*}}}$. Consequently, for stage $r^*$ and all subsequent stages, the allocated budget surpasses the high-probability sample complexity associated with each respective stage.

**Assumption B.2.** The total budget $B$ is large enough such that, for some $\delta_0 < 1$,

$$B \geq 2\left(A\ln\left(\frac{1}{\delta_0}\right) + (C+1)\right)\ln\left(2\frac{A}{Q}\ln\left(\frac{1}{\delta_0}\right) + \frac{2(C+1)}{Q}\right).$$

**Proposition B.3.** *Under Assumption B.2, we have*

$$r^* \leq R$$

*where $R$ is the number of stages (defined in Algorithm 3).*

*Proof.* This is a direct consequence of Lemma K.1. $\qquad\square$

The proposition above establishes that the event in which the allocated budget exceeds the high-probability sample complexity occurs at some stage within $R$, and persists for all subsequent stages thereafter.

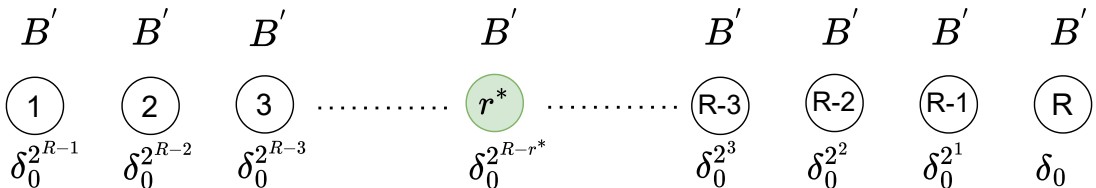

*Figure 3.* Execution routine of Algorithm 3

*Theorem 3.2* (Error probability bound of FC2FB). For a strong fixed confidence algorithm $\mathcal{A}$, under Assumption B.2, for a given budget $B$, with $\delta_0 \leq 0.5$, FC2FB satisfies,

$$\mathbb{P}\left(\hat{J} \neq 1\right) \leq 3\exp\left(-\frac{B}{\frac{4Q}{\ln(\frac{1}{\delta_0})} + 4\log_2\left(\frac{B}{Q}\right)A}\right).$$

*Proof.* Let $\hat{r}$ be the random variable indicating the stage at the end of which the algorithm self-terminates and returns its output. We set $\hat{r} = -1$ if the algorithm does not self-terminate prior to exhausting the budget $B$.

$$\mathbb{P}\left(\hat{J} \neq 1\right) = \mathbb{P}\left(\hat{J} \neq 1, \left((\hat{r} = -1) \vee (\hat{r} > r^*)\right)\right) + \mathbb{P}\left(\hat{J} \neq 1, 1 \leq \hat{r} \leq r^*\right). \qquad\text{(law of total probability)}$$

The first term, where $\left((\hat{r} = -1) \vee (\hat{r} > r^*)\right)$, indicates that stage $r^*$ did not self-terminate (this is true since by Proposition B.3, $r^* \leq R$), despite the fact that the allocated budget of that stage ($B'$) exceeds the $\delta_0^{L_{r^*}}$-stopping time.

$$\mathbb{P}\left(\hat{J} \neq 1, \left((\hat{r} = -1) \vee (\hat{r} > r^*)\right)\right) \leq \mathbb{P}\left((\hat{r} = -1) \vee (\hat{r} > r^*)\right)$$

$$\leq \mathbb{P}\left(\hat{J}_{r^*} = 0\right)$$

$$\leq \delta_0^{L_{r^*}} \qquad \text{(stopping time guarantee, Definition 3.1)}$$

$$= \delta_0^{2^{R-r^*}}$$

$$= \exp\left(-2^{R-r^*} \ln(\frac{1}{\delta_0})\right).$$

Furthermore, if $\hat{r} \leq r^*$, we can leverage the correctness guarantee of $\mathcal{A}$ to establish that the probability of misidentification is correspondingly low due to the small $\delta$ value.

$$\mathbb{P}\left(\hat{J} \neq 1, 1 \leq \hat{r} \leq r^*\right) \leq \mathbb{P}\left(1 \leq \hat{r} \leq r^*\right)$$

$$= \sum_{r=1}^{r^*} \mathbb{P}\left(\hat{J} > 1, \hat{r} = r\right) \qquad \text{(law of total probability)}$$

$$= \sum_{r=1}^{r^*} \mathbb{P}\left(\hat{J}_r > 1\right)$$

$$\leq \sum_{r=1}^{r^*} \delta_0^{2^{R-r}} \qquad \text{(correctness, Definition 3.1)}$$

$$= \delta_0^{2^{R-r^*}} + \delta_0^{2^{R-r^*+1}} + \delta_0^{2^{R-r^*+2}} + \cdots + \delta_0^{2^{R-r^*+(r^*-1)}}$$

$$\leq \delta_0^{2^{R-r^*}} + \delta_0^{2^{R-r^*+1}} + \delta_0^{2^{R-r^*+2}} + \cdots \qquad \text{(infinite sum)}$$

$$\leq \delta_0^{2^{R-r^*}} + \delta_0^{(2^{R-r^*}+1)} + \delta_0^{(2^{R-r^*}+2)} + \cdots \qquad \text{(additional terms)}$$

$$\leq \frac{\delta_0^{2^{R-r^*}}}{1 - \delta_0^{2^{R-r^*}}} \qquad \text{(geometric sum)}$$

$$\leq 2\delta_0^{2^{R-r^*}} \qquad (\delta_0 \leq 0.5)$$

$$= 2\exp\left(-2^{R-r^*} \ln(\frac{1}{\delta_0})\right).$$

It remains to lower bound $R - r^*$.

By Definition B.1 and Proposition B.3,

$$1 \leq r^* \leq R$$

Case 1: $r^* = 1$

$$r^* = 1 \implies R - r^* = R - 1.$$

Thus, we want to bound $R$.

$$R = \lfloor \log_2\left(\frac{B}{Q}\right) \rfloor$$

$$\log_2\left(\frac{B}{Q}\right) \geq R \geq \log_2\left(\frac{B}{Q}\right) - 1$$

$$= \log_2\left(\frac{B}{2Q}\right)$$

$$2^{R-1} \geq \frac{B}{4Q}$$

$$2^{R-r^*} \geq \frac{B}{4Q}.$$

Case 2: $1 < r^* \leq R$

$$B' \leq A \ln \frac{1}{\delta_0^{2^{R-r^*+1}}} + C \qquad \text{(Definition B.1)}$$

$$2^{R-r^*} \geq \frac{B'-C}{2A\ln(\frac{1}{\delta_0})}$$

$$= \frac{\lfloor \frac{B}{R} \rfloor - C}{2A\ln(\frac{1}{\delta_0})}$$

$$\geq \frac{\frac{B}{R} - 1 - C}{2A\ln(\frac{1}{\delta_0})}$$

$$= \frac{B - R(C+1)}{2RA\ln(\frac{1}{\delta_0})}$$

$$\geq \frac{B - R(C+1)}{2\log_2\left(\frac{B}{Q}\right) A\ln(\frac{1}{\delta_0})}$$

$$\geq \frac{B}{4\log_2\left(\frac{B}{Q}\right) A\ln(\frac{1}{\delta_0})}. \qquad (B \geq 2R \cdot (C+1), \text{ Lemma K.1})$$

Hence,

$$2^{R-r^*} \geq \min\left\{\frac{B}{4Q}, \frac{B}{4\log_2\left(\frac{B}{Q}\right) A\ln(\frac{1}{\delta_0})}\right\}$$

$$= \frac{B}{\max\left\{4Q, 4\log_2\left(\frac{B}{Q}\right) A\ln(\frac{1}{\delta_0})\right\}}$$

$$\geq \frac{B}{4Q + 4\log_2\left(\frac{B}{Q}\right) A\ln(\frac{1}{\delta_0})}.$$

Therefore,

$$\mathbb{P}\left(\hat{J} \neq 1\right) \leq 3\exp\left(-2^{R-r^*}\ln(\frac{1}{\delta_0})\right)$$

$$\leq 3\exp\left(-\frac{B}{\frac{4Q}{\ln(\frac{1}{\delta_0})} + 4A\log_2\left(\frac{B}{Q}\right)}\right).$$

This completes the proof. $\qquad \square$

## C. From Weak to Strong Fixed-Confidence Algorithms: FCW2S

Before proceeding to the analysis, we clarify an ambiguity in Algorithm 4. Once an instance $\mathcal{A}_\ell^w$ self-terminates, no further samples are allocated to that instance. The algorithm maintains a set $V \subseteq [L]$, which is initialized as $V \leftarrow \emptyset$, that tracks the indices of terminated trials. This detail is not explicitly stated in the algorithm.

*Proposition 4.2* (Correctness of $FCW2S(\delta)$). For any error rate $\delta \in (0, 1)$, for any weak fixed-confidence algorithm $\mathcal{A}^w$ that satisfies Definition 4.1 with $\delta_0 < \frac{1}{4e}$, running the fixed confidence algorithm FCW2S with $L \geq 4\frac{\ln\frac{1}{\delta}}{\ln\frac{1}{4e\delta_0}}$ satisfies, (1 denotes the optimal arm and $\tau$ denotes the stopping time)

$$\mathbb{P}\left(\hat{J} \neq 1, \tau < \infty\right) < \delta.$$

*Proof.* Let $\mathcal{E}$ be the error event for the FCW2S algorithm, so $\mathcal{E} := \left\{\hat{J} \neq 1, \tau < \infty\right\}$.

By Definition 4.1, for any $\ell \leq L$, the probability of incorrect termination satisfies:

$$\mathbb{P}\left(\hat{J}_\ell > 1\right) \leq \delta_0 .$$ (weak correctness guarantee)

The FCW2S algorithm terminates at time $\tau$, when at least half of the trials have terminated, i.e., $|V| \geq \frac{L}{2}$. It produces an error if, among this set $V$ of terminated trials, the number of incorrect votes is greater than or equal to the number of correct votes. Let $N_V$ be the number of incorrect votes. The condition for an error implies $2N_V \geq |V|$.

Given that the algorithm only stops when $|V| \geq \frac{L}{2}$, a necessary condition for an error to occur is:

$$N_V \geq \frac{|V|}{2} \geq \frac{L}{4} .$$

The set of incorrect votes from the terminated set is a subset of incorrectly terminated trials out of all $L$ trials. Let $N$ be this total count, then $N \geq N_V$. Therefore, a necessary condition for an error is that at least $\frac{L}{4}$ of the total $L$ trials terminated incorrectly. We bound $\mathbb{P}(\mathcal{E})$ as follows:

$$\begin{aligned}
\mathbb{P}(\mathcal{E}) &\leq \mathbb{P}(N \geq \frac{L}{4}) \\
&\leq \exp\left(-\frac{L}{4}\ln\left(\frac{1}{4e\delta_0}\right)\right) && \text{(Lemma K.2, } \delta_0 < \frac{1}{4e}\text{)} \\
&= (4e\delta_0)^{\frac{L}{4}} \\
&\leq \delta. && (L \geq 4\frac{\ln\frac{1}{\delta}}{\ln\frac{1}{4e\delta_0}})
\end{aligned}$$

$\square$

*Proposition 4.3* (Strong stopping time guarantee of $FCW2S(\delta)$). For any error rate $\delta \in (0, 1)$, for any weak fixed-confidence algorithm $\mathcal{A}^w$ that satisfies Definition 4.1 with $\delta_0 < \frac{1}{4e}$, running the fixed confidence algorithm FCW2S with $L \geq 4\frac{\ln\frac{1}{\delta}}{\ln\frac{1}{4e\delta_0}}$ satisfies, ($\tau$ denotes the stopping time)

$$\mathbb{P}\left(\tau > L \cdot f(\delta_0)\right) < \delta.$$

*Proof.* Let $T_\delta^* := L \cdot f(\delta_0)$.

Then, at time $T_\delta^*$, each not-yet-terminated trial $\mathcal{A}_\ell^w$ has been allocated at least $f(\delta_0)$ samples.

By Definition 4.1, for any $\ell \leq L$, the probability a trial has not terminated by $f(\delta_0)$ samples satisfies,

$$\mathbb{P}\left(\hat{J}_\ell = 0\right) \leq \delta_0.$$ (weak stopping time guarantee)

The FCW2S algorithm stops at $\tau \leq T_\delta^*$ iff at least half of the $L$ trials have terminated by time $T_\delta^*$. Therefore, the event $\left\{\tau > T_\delta^*\right\}$ is the same as at time $T_\delta^*$, at least half of the trials are still running. Let $S$ be the total number of learners who have not terminated by $T_\delta^*$. We bound $\mathbb{P}(\tau > T_\delta^*)$ as follows:

$$\mathbb{P}(\tau > T_\delta^*) = \mathbb{P}(S \geq \frac{L}{2})$$

$$\leq \exp\left(-\frac{L}{2}\ln\left(\frac{1}{2e\delta_0}\right)\right) \qquad\qquad \text{(Lemma K.2, } \delta_0 < \frac{1}{4e}\text{)}$$

$$= (2e\delta_0)^{\frac{L}{2}}$$

$$\leq \delta. \qquad\qquad (L \geq 4\frac{\ln\frac{1}{\delta}}{\ln\frac{1}{4e\delta_0}})$$

$\square$

## D. A Meta-Algorithm for Anytime Conversion: FC2AT

In this section, we present an anytime meta-algorithm that builds upon FC2FB using the doubling trick. Algorithm 6 (FC2AT) invokes FC2FB in phases with exponentially increasing budgets allocated to each phase. This algorithm does not require knowledge of the total budget and provides an exponential error probability bound.

---

**Algorithm 6** FC2AT

---

**Input:** Algorithm $\mathcal{A}$, base failure rate $\delta_0$, base budget $Q$
$\hat{J}_0 \leftarrow$ any arbitrary arm, $t \leftarrow 0$
**for** each phase $i = 1, 2, \cdots$ **do**
   $T_i := 2^i \cdot Q$
   Initialize $\mathcal{A}_i = FC2FB\left(T_i, \delta_0, \mathcal{A}, Q\right)$
   **for** $s = 1, 2, \cdots, T_i$ **do**
      $t \leftarrow t + 1$
      pull arm $I_t$ recommended by $\mathcal{A}_i$ at its local time step $s$.
      **if** $s \neq T_i$ **then**
         set $\hat{J}_t \leftarrow \hat{J}$
      **end if**
   **end for**
   $\hat{J} \leftarrow$ the estimated best arm from $\mathcal{A}_i$: $\hat{J}_i$
   $\hat{J}_t \leftarrow \hat{J}$
**end for**

---

**Definition D.1.** Let $B^* := \max\left\{2\left(A\ln\left(\frac{1}{\delta_0}\right) + (C+1)\right)\ln\left(2\frac{A}{Q}\ln\left(\frac{1}{\delta_0}\right) + \frac{2(C+1)}{Q}\right), 2Q\right\}$. Define the optimal phase as

$$i^* := \min\left\{i \geq 1 : T_i \geq B^*\right\}. \qquad\qquad \text{(where } T_i = 2^i \cdot Q\text{)}$$

Index $i^*$ is the phase in which the allocated budget for a phase exceeds the budget requirement of FC2FB instance, as specified in Theorem 3.2.

**Definition D.2.** For any $T \geq \max\left\{(4B^* - 2Q), 2Q\right\}$, define the final phase as

$$I_f := \max\left\{k \geq 1 : \sum_{i=1}^{k} T_i \leq T\right\}. \qquad\qquad \text{(where } T_i = 2^i \cdot Q\text{)}$$

For any time $T \geq \max\left\{(4B^* - 2Q), 2Q\right\}$, $I_f$ denote the index of the last phase that completed. Since $T \geq 2Q \implies T \geq T_1$, $I_f$ is well defined.

**Proposition D.3.** *For any* $T \geq \max\left\{(4B^* - 2Q), 2Q\right\}$,

$$T_{I_f} \geq B^*. \qquad\qquad \text{(where } T_{I_f} = 2^{I_f} \cdot Q\text{)}$$

*Proof.*

$$\sum_{i=1}^{i^*} T_i = \frac{T_1 \cdot 2^{i^*} - T_1}{2 - 1}$$
$$= 2Q \cdot 2^{i^*} - 2Q$$
$$= 4Q \cdot 2^{i^*-1} - 2Q.$$

Case 1: $i^* = 1$,

$$\sum_{i=1}^{i^*} T_i = 4Q \cdot 2^{i^*-1} - 2Q$$
$$= 4Q - 2Q$$
$$= 2Q$$
$$\leq T.$$

Case 2: $i^* > 1$,

$$\sum_{i=1}^{i^*} T_i = 4Q \cdot 2^{i^*-1} - 2Q$$
$$= 4T_{i^*-1} - 2Q$$
$$\leq 4B^* - 2Q \qquad \text{(Definition D.1)}$$
$$\leq T.$$

Hence,

$$T \geq \sum_{i=1}^{i^*} T_i.$$

Therefore,

$$I_f \geq i^*. \tag{4}$$

Subsequently,

$$T_{I_f} \geq B^*. \qquad \text{(Definition D.1)}$$

$\square$

**Proposition D.4.** *For any $T \geq \max\left\{ (4B^* - 2Q), 2Q \right\}$,*

$$T_{I_f} \geq \frac{T}{4}.$$

*Proof.*

$$\sum_{i=1}^{I_f+1} T_i \geq T$$
$$\Rightarrow \frac{2^{I_f+1} - 1}{2 - 1} T_1 \geq T$$
$$\Rightarrow 2^{I_f+2} Q - 2 \cdot Q \geq T$$
$$\Rightarrow 2^{I_f+2} Q \geq T$$
$$\Rightarrow I_f \geq \log\left(\frac{T}{4 \cdot Q}\right).$$

Therefore,

$$T_{I_f} = 2^{I_f} \cdot Q \geq \frac{T}{4}.$$

$\square$

**Theorem D.5** (Error probability bound for FC2AT). *For a strong fixed-confidence algorithm $\mathcal{A}$, let $T \geq \max\left\{(4B^* - 2Q), 2Q\right\}$, FC2AT satisfies,*

$$\mathbb{P}\left(\hat{J}_T \neq 1\right) \leq 3 \exp\left(-\frac{T}{\frac{16Q}{\ln(\frac{1}{\delta_0})} + 16 \log_2\left(\frac{T}{Q}\right) A}\right).$$

*Proof.*

$$\mathbb{P}\left(\hat{J}_T \neq 1\right) = \mathbb{P}\left(\hat{J}_{I_f} \neq 1\right)$$

$$\leq 3 \cdot \exp\left(-\frac{T_{I_f}}{\frac{4Q}{\ln(\frac{1}{\delta_0})} + 4 \log_2\left(\frac{T_{I_f}}{Q}\right) A}\right) \qquad \text{(Theorem 3.2 and Proposition D.3: } T_{I_f} \geq B^*\text{)}$$

$$\leq 3 \exp\left(-\frac{T_{I_f}}{\frac{4Q}{\ln(\frac{1}{\delta_0})} + 4 \log_2\left(\frac{T}{Q}\right) A}\right)$$

$$\leq 3 \exp\left(-\frac{T}{\frac{16Q}{\ln(\frac{1}{\delta_0})} + 16 \log_2\left(\frac{T}{Q}\right) A}\right). \qquad \text{(Proposition D.4)}$$

This completes the proof. $\square$

# E. Strong and Weak Fixed-Confidence Algorithms and Their Relation to Existing Methods

In this section, we introduce strong and weak fixed-confidence BAI algorithms and explain how they relate to fixed-confidence algorithms with guarantees on high-probability sample complexity, expected sample complexity, asymptotic sample complexity, and exponentially decaying stopping times. Throughout this section, the correctness guarantee is uniform over the model class, while sample-complexity quantities such as $f(\cdot)$, $A$, and $C$ may depend on the underlying instance $\nu$, but not on the target confidence level $\delta$, unless explicitly stated otherwise.

*Property* 1 (High probability sample complexity bound). An FC algorithm is said to enjoy a high-probability sample complexity bound if for any $\delta \in (0, 1)$, it satisfies $\delta$-correctness (see (1)) and has a high probability sample complexity of $T_\delta^* = f(\delta)$ (see (2)), where $f(\delta)$ denotes some function of $\delta$ that does not necessarily have logarithmic dependence on $1/\delta$.

*Property* 2 (Weak sample complexity bound). An FC algorithm is said to enjoy a high-probability sample complexity bound if for **some** $\delta_0 \in (0, 1)$, it satisfies $\delta_0$-correctness (see (1)) and has a high probability sample complexity of $T_{\delta_0}^* = f(\delta_0)$ (see (2)), where $f(\delta_0)$ denotes some function of $\delta_0$ that does not necessarily have logarithmic dependence on $1/\delta_0$.

All algorithms with weak sample complexity bound are Weak FC algorithms (Definition 4.1).

*Property* 3 (Strong sample complexity bound). An FC algorithm is said to enjoy a Strong sample complexity bound if for any $\delta \in (0, 1)$, it satisfies $\delta$-correctness (see (1)) and has a high probability sample complexity of $T_\delta^* = A \ln(1/\delta) + C$ (see (2)) for some $A$ and $C$ independent of $\delta$.

All algorithms with Strong sample complexity bound are Strong FC algorithms (Definition 3.1).

*Property* 4 (Expected sample complexity bound). An FC algorithm is said to enjoy an expected sample complexity bound if for any $\delta \in (0, 1)$, it satisfies $\delta$-correctness (see (1)) and

$$\mathbb{E}[\tau] \leq T_\delta^*,$$

where $\tau$ (stopping time) depends on $\delta$.

*Property* 5 (Asymptotic expected sample complexity). An FC algorithm is said to enjoy an asymptotic expected sample complexity of $A$, if for any $\delta \in (0, 1)$, it satisfies $\delta$-correctness (see (1)) and

$$\limsup_{\delta \to 0} \frac{\mathbb{E}[\tau]}{\ln(1/\delta)} \leq A \, ,$$

where $\tau$ (stopping time) depends on $\delta$ and $A$ is independent of $\delta$.

*Property* 6 (Exponential stopping tail). A fixed confidence algorithm is said to have a $(T_\delta, \kappa)$-exponential stopping tail if for any $\delta \in (0, 1)$ it satisfies $\delta$-correctness and there exists a time step $T_\delta^*$ and a problem-dependent constant $\kappa > 0$ (but not dependent on $T$) such that for all $T \geq T_\delta^*$,

$$\mathbb{P}\left(\tau \geq T\right) \leq \exp\left(-\frac{T}{\kappa \cdot \mathrm{polylog}(T)}\right) .$$

where $\tau$ (stopping time) depends on $\delta$. See Definition 2.8 of Balagopalan et al. (2025).

The following claim holds,

*Claim* 1. Any algorithm that enjoys either High probability sample complexity bound, Strong sample complexity bound, Expected sample complexity bound, or Exponential stopping tail, also enjoys Weak probability sample complexity bound.

1. Property 1 $\implies$ Property 2

   This hold by definition.

2. Property 3 $\implies$ Property 1, 2

   This hold by definition.

3. Property 4 $\implies$ Property 1, 2

   By Markov's inequality,

   $$\mathbb{P}\left(\tau \geq T\right) \leq \frac{\mathbb{E}[\tau]}{T} \leq \frac{T_\delta^*}{T} \overset{\overset{\text{set to}}{\frown}}{=} \delta'$$

   Hence, for any $\delta' \in (0, 1)$

   Let

   $$T_{\delta'}^* = \frac{T_\delta^*}{\delta'} = f(\delta')$$

   Therefore,

   $$\mathbb{P}\left(\tau \geq T_{\delta'}^*\right) \leq \delta'$$

   Thus it satisfies Property 1 and Property 2 follows.

4. Property 6 $\implies$ Property 1, 2, 4.

   See (Balagopalan et al., 2025).

To help with the understanding, we now discuss a few representative algorithms from the literature.

- Successive Elimination ((Even-Dar et al., 2006)) enjoys Property 1 and Property 3, but not Property 4 and Property 6 (see Table 1 from Balagopalan et al. (2025)).

- LUCB1 ((Kalyanakrishnan et al., 2012)) enjoys Property 1, Property 3 and Property 4, but whether it enjoys Property 6 is unknown (see Table 1 from Balagopalan et al. (2025)).

- Uniform sampling enjoys Property 1 and Property 3, Property 4 and Property 6. However, for uniform sampling, these bounds are not optimal in general.

Since we have presented the framework FCW2S (Algorithm 4) to convert any Weak FC algorithm into a Strong FC algorithm, we can construct a fixed-budget algorithm by applying FC2FB (Algorithm 3) framework on most of the fixed-confidence algorithms in literature and theoretically analyze them. Moreover, we can still apply FC2FB on fixed-confidence algorithms that are not Weak FC (Eg: Track-and-Stop ((Garivier & Kaufmann, 2016))). However, the analysis of the latter framework is an open problem.

## F. A Candidate for Strong Fixed-Confidence Algorithms in the Heterogeneous Noise Setting: PE-KHN

In this section, we prove Theorem 5.1 for Algorithm 5. The algorithm is an adaptation of the elimination algorithm for pure exploration presented in Section 1.2 of Jamieson (2021), extended to the heterogeneous noise case with the additional enhancement of accumulating samples across stages.

**Definition F.1.** Define $\Delta_j = \mu_1 - \mu_j \ \ \forall j \in [K], j \neq 1$.

**Definition F.2.** Define

$$\mathcal{E}_{1,\ell} = \left\{ \mu_1 - \hat{\mu}_1^{(\ell)} \leq \varepsilon_\ell \right\}, \text{ (where } \hat{\mu}_1^\ell \text{ denotes the average reward of arm 1 across all stages up to and including stage } \ell.)$$

and, $\forall i \in [K], i \neq 1$,

$$\mathcal{E}_{i,\ell} = \left\{ \hat{\mu}_i^{(\ell)} - \mu_i \leq \varepsilon_\ell \right\}. \text{ (where } \hat{\mu}_i^\ell \text{ denotes the average reward of arm } i \text{ across all stages up to and including stage } \ell.)$$

Let $n_i^{(\ell)}$ denote the number of times arm $i$ has been pulled up to and including stage $\ell$.

We can assume that the sequence of rewards for each arm is drawn before the beginning of the game. Thus the empirical reward for arm $i$ after $n_i^{(\ell)}$ pulls is well defined even if arm $i$ has not been actually pulled $n_i^{(\ell)}$ times. This assumption, consistent with that of Audibert et al. (2010), not only facilitates the accessibility of the subsequent proof but also establishes the independence between samples.

By Hoeffding's inequality for sub-Gaussian random variables, $\forall i \in [K]$,

$$
\begin{aligned}
\mathbb{P}\left(\mathcal{E}_{i,\ell}^c\right) &\leq \exp\left(-\frac{n_i^{(\ell)} \varepsilon_\ell^2}{2\sigma_i^2}\right) \\
&\leq \exp\left(-\frac{\frac{2\sigma_i^2}{(\varepsilon_\ell)^2}\ln\left(K/\delta_\ell\right)\varepsilon_\ell^2}{2\sigma_i^2}\right) \quad\quad \text{(by definition of Algorithm 5)} \\
&= \frac{\delta_l}{K} \\
&= \frac{\delta}{\ell(\ell+1)K}.
\end{aligned}
$$

Hence,

$$\mathbb{P}\left(\mathcal{E}_{i,\ell}^c\right) \leq \frac{\delta}{\ell(\ell+1)K}.$$

Let us define

**Definition F.3.** Define,

$$\mathcal{E} = \bigcap_{i=1}^{K} \bigcap_{\ell=1}^{\infty} \mathcal{E}_{i,\ell}.$$

Hence,

$$\mathcal{E}^c = \bigcup_{i=1}^{K} \bigcup_{\ell=1}^{\infty} \mathcal{E}_{i,\ell}^c.$$

$$\mathbb{P}\left(\mathcal{E}^c\right) = \mathbb{P}\left( \bigcup_{i=1}^{K} \bigcup_{\ell=1}^{\infty} \mathcal{E}_{i,\ell}^c \right)$$

$$\leq \sum_{i=1}^{K} \sum_{\ell=1}^{\infty} \mathbb{P}\left(\mathcal{E}_{i,\ell}^c\right)$$

$$\leq \sum_{i=1}^{K} \sum_{\ell=1}^{\infty} \frac{\delta}{\ell(\ell+1)K}$$

$$= \sum_{\ell=1}^{\infty} \frac{\delta}{\ell(\ell+1)}$$

$$= \delta.$$

For the remainder of the proof, assume that $\mathcal{E}$ holds.

**Proposition F.4** (Optimal arm survives). *If $\mathcal{E}$ holds then, the optimal arm will never be eliminated.*
$$1 \in \mathcal{S}_\ell \quad \forall \ell.$$

*Proof.* Fix any $\ell$ for which $1 \in \mathcal{S}_\ell$.

For any $j \in \mathcal{S}_{\ell+1}, j \neq 1$ we have,

$$\hat{\mu}_j^{(\ell)} - \hat{\mu}_1^{(\ell)} \leq \left(\hat{\mu}_j^{(\ell)} - \hat{\mu}_1^{(\ell)}\right) + \left(\mu_1 - \mu_j\right) \qquad \text{(because } \left(\mu_1 - \mu_j\right) \geq 1\text{)}$$

$$= \left(\hat{\mu}_j^{(\ell)} - \mu_j\right) + \left(\mu_1 - \hat{\mu}_1^{(\ell)}\right)$$

$$\leq 2\varepsilon_\ell. \qquad \text{(because } \mathcal{E} \text{ holds)}$$

Since the above statement is true for any $j \in \mathcal{S}_{\ell+1}$,
$$\max_{j \in \mathcal{S}_{\ell+1}} \hat{\mu}_j^{(\ell)} - \hat{\mu}_1^{(\ell)} \leq 2\varepsilon_\ell.$$

Hence,

$$1 \in \mathcal{S}_{\ell+1}.$$

Since $1 \in \mathcal{S}_1$, the optimal arm will never be eliminated. This leads to the following theorem.

$\square$

**Theorem F.5** (Correctness of PE-KHN). *For any $\delta \leq 1$, let J be the output of Algorithm 5 and 1 be the optimal arm, then*
$$\mathbb{P}(\hat{J} \neq 1) \leq \delta.$$

**Proposition F.6** (Near-optimal arms survive). *If $\mathcal{E}$ holds then, and $j \in \mathcal{S}_{\ell+1}$ then,*
$$\Delta_j \leq 8\varepsilon_{\ell+1}.$$

*Proof.* For any $j \in S_{\ell+1}$

$$\max_{i \in S_{\ell+1}} \hat{\mu}_i^{(\ell)} - \hat{\mu}_j^{(\ell)} \leq 2\varepsilon_\ell.$$

Also, since $1 \in S_{\ell+1}$

$$\max_{i \in S_{\ell+1}} \hat{\mu}_i^{(\ell)} - \hat{\mu}_1^{(\ell)} \geq 0.$$

Combining both,

$$\hat{\mu}_1^{(\ell)} - \hat{\mu}_j^{(\ell)} \leq 2\varepsilon_\ell$$
$$\hat{\mu}_1^{(\ell)} - \hat{\mu}_j^{(\ell)} + \Delta_j \leq 2\varepsilon_\ell + \left(\mu_1 - \mu_j\right)$$
$$\Delta_j \leq 2\varepsilon_\ell + \left(\hat{\mu}_j^{(\ell)} - \mu_j\right) + \left(\mu_1 - \hat{\mu}_1^{(\ell)}\right)$$
$$\leq 4\varepsilon_\ell \qquad\qquad \text{(because } \mathcal{E} \text{ holds)}$$
$$= 8\varepsilon_{\ell+1}.$$

$\square$

**Proposition F.7** (Elimination stages). *If $\mathcal{E}$ holds, then the elimination stage of arm $j \neq 1$ is given by,*
$$\ell_j^* \leq \lceil \log_2\left(4\Delta_j^{-1}\right)\rceil.$$

*Proof.* If $\mathcal{E}$ holds then, by Proposition F.6, when an arm $j$ is getting eliminated,
$$\Delta_j > 8\varepsilon_{\ell+1} \left(= \frac{8}{2^{\ell+1}}\right) \implies j \notin \mathcal{S}_{\ell+1}$$
$$\ell > \log_2\left(4\Delta_j^{-1}\right) \implies j \notin \mathcal{S}_{\ell+1}.$$
Hence, if $\mathcal{E}$ holds then, the maximum stage that a sub-optimal arm $j$ can survive is,
$$\ell_j^* \leq \lceil \log_2\left(4\Delta_j^{-1}\right)\rceil$$

$\square$

*Theorem 5.1* (Sample complexity of PE-KHN). For any $\delta \leq 1$, let $T_\delta^* := \frac{64\sigma_1^2}{\Delta^2} \ln\left(\frac{4K(\ln 2)^2}{\delta}\left(\ln\left(\frac{4}{\Delta}\right)\right)^2\right) +$
$\sum_{j\neq 1}^{K} \frac{64\sigma_j^2}{\Delta_j^2} \ln\left(\frac{4K(\ln 2)^2}{\delta}\left(\ln\left(\frac{4}{\Delta_j}\right)\right)^2\right)$, assume $\Delta_j \leq 1 \ \forall j \neq 1 \in \mathcal{S}$, then
$$\mathbb{P}(\tau > T_\delta^*) \leq \delta.$$

*Proof.* If $\mathcal{E}$ holds, tthen by Proposition F.7, the total samples consumed by arm $j$ up to its elimination is

$$T_j = \lceil \frac{2\sigma_j^2}{(\varepsilon_{\ell_j^*})^2} \ln(K/\delta_{\ell_j^*})\rceil$$
$$\leq \frac{2\sigma_j^2}{(\varepsilon_{\ell_j^*})^2} \ln(K/\delta_{\ell_j^*}) + 1$$
$$\leq 2\sigma_j^2 2^{2\log_2\left(4\Delta_j^{-1}\right)+2} \ln\left(\frac{K\left(\log_2\left(4\Delta_j^{-1}\right)+1\right)\left(\log_2\left(4\Delta_j^{-1}\right)+2\right)}{\delta}\right)$$
$$= \frac{64\sigma_j^2}{\Delta_j^2} \ln\left(\frac{K\left(\log_2\left(4\Delta_j^{-1}\right)+1\right)\left(\log_2\left(4\Delta_j^{-1}\right)+2\right)}{\delta}\right)$$

$$\leq \frac{64\sigma_j^2}{\Delta_j^2} \ln \left( \frac{K \left( 2\log_2 \left( 4\Delta_j^{-1} \right) \right) \left( 2\log_2 \left( 4\Delta_j^{-1} \right) \right)}{\delta} \right) \qquad \text{(assume } \forall j \ \Delta_j < 1\text{)}$$

$$= \frac{64\sigma_j^2}{\Delta_j^2} \ln \left( \frac{4K \left( \log_2 \left( 4\Delta_j^{-1} \right) \right)^2}{\delta} \right)$$

$$= \frac{64\sigma_j^2}{\Delta_j^2} \ln \left( \frac{4K(\ln 2)^2}{\delta} \left( \ln \left( \frac{4}{\Delta_j} \right) \right)^2 \right).$$

Hence the stopping time $\tau$ will be upper bounded by

$$\tau \leq \sum_{j \neq 1}^{K} \frac{64\sigma_j^2}{\Delta_j^2} \ln \left( \frac{4K(\ln 2)^2}{\delta} \left( \ln \left( \frac{4}{\Delta_j} \right) \right)^2 \right) + T_1.$$

(where $T_1$ in the number of times arm 1 chosen up to that point)

Given the $\mathcal{E}$ holds, the arm 1 will survive until the second best arm is eliminated.

$$\ell_2 \leq \lceil \log_2 \left( 4\Delta_2^{-1} \right) \rceil.$$

$$T_1 \leq \lceil \frac{2\sigma_1^2}{(\varepsilon_\ell)^2} \ln(K/\delta_\ell) \rceil$$

$$\leq \frac{64\sigma_1^2}{\Delta_2^2} \ln \left( \frac{4K(\ln 2)^2}{\delta} \left( \ln \left( \frac{4}{\Delta_2} \right) \right)^2 \right).$$

Hence the stopping time $\tau$ will be upper bounded by

$$\tau \leq \frac{64\sigma_1^2}{\Delta_2^2} \ln \left( \frac{4K(\ln 2)^2}{\delta} \left( \ln \left( \frac{4}{\Delta_2} \right) \right)^2 \right) + \sum_{j \neq 1}^{K} \frac{64\sigma_j^2}{\Delta_j^2} \ln \left( \frac{4K(\ln 2)^2}{\delta} \left( \ln \left( \frac{4}{\Delta_j} \right) \right)^2 \right)$$

$$= \mathcal{O}\left( A\log\left( \frac{1}{\delta} \right) \right). \qquad (A = \mathcal{O}\left( \frac{\sigma_1}{\Delta_2^2} + \sum_{i \in S_1, i \neq 1} \frac{\sigma_i}{\Delta_i^2} \right), \text{ ignoring doubly logarithmic factors and } \log K)$$

This completes the proof. $\qquad \square$

## G. Linear Bandits

In this section, we use $A \lesssim B$ to denote that $A \leq c \cdot B$ for some numerical constant $c$. Recall that the optimal sample complexity in FC is governed by $\rho^*$ defined in Fiez et al. (2019):

$$\rho^* = \min_{\lambda \in \Delta(\mathcal{X})} \max_{x \in \mathcal{X} \backslash \{x^*\}} \frac{\|x^* - x\|_{(\sum_{x \in \mathcal{X}} \lambda_x x x^\top)^{-1}}^2}{\langle x^* - x, \theta \rangle^2}.$$

On the other hand, in FB, Yang & Tan (2022) showed that it is possible to obtain a sample complexity bound for which the leading term has the following problem-dependent constant:

$$H_{2,\text{lin}} = \max_{2 \leq i \leq d} \frac{i}{\Delta_i^2}.$$

**Theorem G.1.** *We have $\rho^* \lesssim H_{2,lin} \ln^2(d)$.*

*Proof.* Let $\mathcal{X} \subset \mathbb{R}^d$ be the set of arms spanning $\mathbb{R}^d$, and $x^*$ be the unique optimal arm for a parameter $\theta^* \in \mathbb{R}^d$. The Dirac delta measure, $\delta_x$, is a distribution with its entire mass on arm $x$. A design $\lambda$ is a probability distribution on $\mathcal{X}$, and the design matrix is $V(\lambda) = \sum \lambda_x x x^\top$. Its pseudoinverse is denoted $V(\lambda)^\dagger$. The performance gap for an arm $x$ is $\Delta_x = \langle x^* - x, \theta^* \rangle$. We partition arms into nested sets $\mathcal{X}_\ell = \{x : \Delta_x \le \Delta_{(d_\ell)}\}$ based on the $i$-th smallest gap $\Delta_{(i)}$ and sizes $d_\ell = \lceil d \cdot 2^{-\ell+1} \rceil$, where $L$ is the smallest integer such that $d_{L+1} = 1$.

We have

$$
\begin{aligned}
\rho^* &= \min_{\lambda \in \Delta(\mathcal{X})} \max_{x \in \mathcal{X}^-} \frac{\|x^* - x\|^2_{V(\lambda)^\dagger}}{\Delta_x^2} \\
&\le \min_{\lambda \in \Delta(\mathcal{X})} \sum_{\ell=1}^{L+1} \max_{x \in \mathcal{X}_{\ell-1}^-} \frac{\|x^* - x\|^2_{V(\lambda)^\dagger}}{\Delta_x^2} \\
&\le \min_{\{\lambda_\ell\}_{\ell=1}^{L+1}} \sum_{\ell=1}^{L+1} \max_{x \in \mathcal{X}_{\ell-1}^-} \frac{\|x^* - x\|^2_{V(\frac{1}{2}\delta_{x^*} + \frac{1}{2(L+1)}\sum_{\ell'} \lambda_{\ell'})^\dagger}}{\Delta_x^2} \\
&\le \min_{\{\lambda_\ell\}_{\ell=1}^{L+1}} \sum_{\ell=1}^{L+1} \max_{x \in \mathcal{X}_{\ell-1}^-} \frac{\|x^* - x\|^2_{V(\frac{1}{2}\delta_{x^*} + \frac{1}{2(L+1)}\lambda_\ell)^\dagger}}{\Delta_x^2} \\
&\le \sum_{\ell=1}^{L+1} \min_{\lambda \in \Delta(\mathcal{X}_{\ell-1}^-)} \max_{x \in \mathcal{X}_{\ell-1}^-} \frac{\|x^* - x\|^2_{V(\frac{1}{2}\delta_{x^*} + \frac{1}{2(L+1)}\lambda)^\dagger}}{\Delta_x^2}.
\end{aligned}
$$

Let $T_\ell$ denote the $\ell$-th term in the final summation.

$$
\begin{aligned}
T_\ell &= \min_{\lambda \in \Delta(\mathcal{X}_{\ell-1}^-)} \max_{x \in \mathcal{X}_{\ell-1}^-} \frac{\|x^* - x\|^2_{V(\frac{1}{2}\delta_{x^*} + \frac{1}{2(L+1)}\lambda)^\dagger}}{\Delta_x^2} \\
&\le \frac{1}{\Delta_{(d_{\ell-1})}^2} \min_{\lambda \in \Delta(\mathcal{X}_{\ell-1}^-)} \max_{x \in \mathcal{X}_{\ell-1}^-} \|x^* - x\|^2_{V(\frac{1}{2}\delta_{x^*} + \frac{1}{2(L+1)}\lambda)^\dagger} \\
&\le \frac{1}{\Delta_{(d_{\ell-1})}^2} \left(4 + 4(L+1) \min_{\lambda \in \Delta(\mathcal{X}_{\ell-1}^-)} \max_{x \in \mathcal{X}_{\ell-1}^-} \|x\|^2_{V(\lambda)^\dagger}\right).
\end{aligned}
$$

To bound the final $\min\max$ term, where $V(\lambda)$ is singular, we project the problem. Let $\mathcal{S}_\ell = \mathrm{span}(\mathcal{X}_{\ell-1}^-)$ be the subspace of dimension $k_\ell \le d_{\ell-1} - 1$. Let the columns of $U_\ell \in \mathbb{R}^{d \times k_\ell}$ form an orthonormal basis for $\mathcal{S}_\ell$. For any $x \in \mathcal{S}_\ell$, the norm can be rewritten by projecting into the subspace:

$$
\|x\|^2_{V(\lambda)^\dagger} = x^\top V(\lambda)^\dagger x = (U_\ell^\top x)^\top (U_\ell^\top V(\lambda) U_\ell)^{-1} (U_\ell^\top x).
$$

Let $x' = U_\ell^\top x$ be the projected vector in $\mathbb{R}^{k_\ell}$. Let $V'(\lambda) = U_\ell^\top V(\lambda) U_\ell$ be the invertible $k_\ell \times k_\ell$ design matrix in the subspace. The problem becomes:

$$
\min_{\lambda \in \Delta(\mathcal{X}_{\ell-1}^-)} \max_{x \in \mathcal{X}_{\ell-1}^-} \|x\|^2_{V(\lambda)^\dagger} = \min_{\lambda \in \Delta(\mathcal{X}_{\ell-1}^-)} \max_{x' \in \{U_\ell^\top x | x \in \mathcal{X}_{\ell-1}^-\}} \|x'\|^2_{V'(\lambda)^{-1}}.
$$

By the Kiefer-Wolfowitz theorem in this $k_\ell$-dimensional space,

$$
\min_{\lambda \in \Delta(\mathcal{X}_{\ell-1}^-)} \max_{x' \in \{U_\ell^\top x | x \in \mathcal{X}_{\ell-1}^-\}} \|x'\|^2_{V'(\lambda)^{-1}} \le k_\ell \le d_{\ell-1} - 1.
$$

Substituting this back into $T_\ell$:

$$
T_\ell \le \frac{4 + 4(L+1)(d_{\ell-1} - 1)}{\Delta_{(d_{\ell-1})}^2} \lesssim \frac{\ln(d) \cdot d_{\ell-1}}{\Delta_{(d_{\ell-1})}^2}.
$$

Summing the bounds for all terms, we have

$$
\rho^* \le \sum_{\ell=1}^{L+1} T_\ell \lesssim \ln(d) \sum_{\ell=1}^{L+1} \frac{d_{\ell-1}}{\Delta_{(d_{\ell-1})}^2} \lesssim \ln(d)^2 H_{2,\text{lin}}.
$$

$\square$

In the following proposition, we show that the ratio between the two sample complexity measures can be made arbitrarily

large.

**Proposition G.2.** *There exists a set of problem instances parametrized by $\varepsilon > 0$ where the sample complexity measures $\rho^*$ from [Fiez et al. (2019)](#) is $\Theta(d)$ but $H_{2,lin}$ from [Yang & Tan (2022)](#) is $\Theta(\frac{d^{1/2}}{\varepsilon^2})$.*

*Proof.* Consider the following instance. Let the true parameter be $\theta = (1, 1, \ldots, 1)^\top$. The top $d$ arms are designed as follows: for $i = 1$,

$$x_1 = (1, 0, \ldots, 0)^\top$$

and for $i = 2, \ldots, d$,

$$x_i = (1 - i^{1/4}\varepsilon, 0, \ldots, 0)^\top.$$

Let the total number of arms be $n = 2d - 1$. The remaining $n - d = d - 1$ arms are designed as

$$x_{d+j} = (0, \ldots, 0, 0.5, 0, \ldots, 0)^\top,$$

where the $0.5$ is at the $(j+1)$-th coordinate, for $j = 1, \ldots, d-1$.

First, we compute $H_{2,\lin}$ for this instance:

$$H_{2,\lin} = \max_{2 \le i \le d} \frac{i}{\Delta_i^2} = \max_{2 \le i \le d} \frac{i}{(i^{1/4}\varepsilon)^2} = \max_{2 \le i \le d} \frac{i^{1/2}}{\varepsilon^2} = \frac{d^{1/2}}{\varepsilon^2}.$$

Next, we analyze $\rho^*$. By definition:

$$
\rho^* = \min_{\lambda \in \Delta(\mathcal{X})} \max_{x \in \mathcal{X} \setminus \{x^*\}} \frac{\|x^* - x\|^2_{(\sum_{x \in \mathcal{X}} \lambda_x x x^\top)^{-1}}}{\langle x^* - x, \theta \rangle^2}
$$

$$
= \min_{\lambda \in \Delta(\mathcal{X})} \max \left\{ \max_{i=2,\ldots,d} \frac{\|(i^{1/4}\varepsilon, 0, \ldots, 0)\|^2_{(\sum_{x \in \mathcal{X}} \lambda_x x x^\top)^{-1}}}{(i^{1/4}\varepsilon)^2}, \max_{j=1,\ldots,d-1} \frac{\|(1, 0, \ldots, -0.5, 0, \ldots, 0)\|^2_{(\sum_{x \in \mathcal{X}} \lambda_x x x^\top)^{-1}}}{0.5^2} \right\}
$$

To bound this quantity, consider a uniform allocation $\lambda'$ over the arms $\{x_1\} \cup \{x_i\}_{i=d+1}^n$. This choice provides an upper bound on $\rho^*$:

$$
\rho^* \le \max \left\{ \max_{i=2,\ldots,d} \frac{\|(i^{1/4}\varepsilon, 0, \ldots, 0)\|^2_{(H')^{-1}}}{(i^{1/4}\varepsilon)^2}, \max_{j=1,\ldots,d-1} \frac{\|(1, 0, \ldots, -0.5, 0, \ldots, 0)\|^2_{(H')^{-1}}}{0.25} \right\}
$$

where the design matrix $H'$ under this allocation is:

$$
H' = \frac{1}{d} x_1 x_1^\top + \frac{1}{d} \sum_{i=d+1}^n x_i x_i^\top = \begin{pmatrix} \frac{1}{d} & 0 & 0 & \cdots & 0 \\ 0 & \frac{1}{2d} & 0 & \cdots & 0 \\ 0 & 0 & \frac{1}{2d} & \cdots & 0 \\ \vdots & \vdots & \vdots & \ddots & \vdots \\ 0 & 0 & 0 & \cdots & \frac{1}{2d} \end{pmatrix}
$$

This leads to the bound:

$$
\rho^* \le \max \left\{ \frac{(i^{1/2}\varepsilon^2)d}{i^{1/2}\varepsilon^2}, \frac{1^2 \cdot d + (-0.5)^2 \cdot 4d}{0.25} \right\} = \max \left\{ d, \frac{2d}{0.25} \right\} = 8d = \Theta(d).
$$

Therefore, for a sufficiently small $\varepsilon$, we have established the separation:

$$
H_{2,\lin} = \frac{d^{1/2}}{\varepsilon^2} \gg \Theta(d) \ge \rho^*.
$$

$\square$

## H. Unimodal Bandits

*Proposition 5.8.* Suppose the unimodal means lies in $[0, 1]$, then

$$T_\mu(\delta_1) + K = \tilde{O}(\Delta^{-2})$$

and there exists an instance where the above two quantities are order-wise different, i.e., $T_\mu(\delta_1) + K = o(\Delta^{-2})$. The quantity $T_\mu(\delta)$ is defined in their Theorem 3.7 in [Poiani et al. (2024)](#) and $\Delta := \min_{2 \le i \le K} |\mu_i - \mu_{i-1}|$ as defined in (Ghosh

et al., 2024).

*Proof.* We will use notations $C_{\mu,1}, T^*_{1/2}(\mu), T_0(\delta), \tilde{T}_0(\delta)$ defined in Theorem 3.7 in Poiani et al. (2024). Taking $\varepsilon = 1$ and $\gamma = a_\mu/2$, one has $C_{\mu,1} = O(H_1 \log H_1)$, $T^*_{1/2}(\mu) = O(H_1)$, where $H_1 = \sum_{i \neq i^*} \Delta_i^{-2}$ and $\Delta_i = \mu_{i^*} - \mu_i$, where $i^* = \arg\max_{i \in [K]} \mu_i$.

If the best arm has one neighbor, $T_\mu(\delta_1) \leq \max(T_0(\delta_1), C_{\mu,1}) = \max(O(T^*_{1/2}(\mu)), C_{\mu,1}) = O(H_1 \log H_1)$.

If the best arm has two neighbors, by the second part of Theorem 3.7, $T_\mu(\delta_1) \leq \max(\tilde{T}_0(\delta_1), C_{\mu,1}) = \max(O(T^*_{1/2}(\mu)), C_{\mu,1}) = O(H_1 \log H_1)$.

In both cases, $T_\mu(\delta_1) = O(H_1 \log H_1)$. As the unimodal means lies in $[0,1]$, $T_\mu(\delta_1) + K = O(H_1 \log H_1)$.

Furthermore, $H_1 = \sum_{i \neq i^*} \Delta_i^{-2} \leq \sum_{i \neq i^*} ((i - i^*)\Delta)^{-2} \leq O(\Delta^{-2})$.

To construct an instance where the quantities $H_1$ and $\Delta^{-2}$ are order-wise different, consider the unimodal means are $(0, \varepsilon, \varepsilon + \frac{1}{2}, \varepsilon, 0)$ for some small $\varepsilon$. In this case, $H_1 = \sum_{i \neq i^*} \Delta_i^{-2} = O(1)$, whereas $\Delta^{-2} = \frac{1}{\varepsilon^2}$. □

## I. Cascading Bandits

In cascading bandits, which model user interactions in recommender systems under a sequential click process, the best-arm identification problem has so far only been studied under the fixed-confidence formulation. Zhong et al. (2020) proposed CASCADEBAI, the first algorithm with theoretical guarantees in this setting, and established upper and lower bounds on its sample complexity. To date, no theoretical result is known for the fixed-budget counterpart. This leaves open an important gap: fixed-budget algorithms are particularly relevant when the exploration horizon is limited by design constraints, as in recommender systems with fixed user engagement.

Let the ground set of items be $[L] := \{1, \ldots, L\}$. Each item $i \in [L]$ has an unknown click probability $w(i) \in [0,1]$. An arm is a list of $K$ items: $S = (i_1, \ldots, i_K) \in [L]^{(K)}$. At each round $t$, the agent plays an arm $S_t$ and receives cascading feedback: the user examines items sequentially until clicking on one (if any).

Feedback on item $i$ at time $t$ is $W_t(i) \sim \text{Bern}(w(i))$, i.i.d. across time. The feedback vector is $O_t \in \{0, 1, ?\}^K$, where "?" denotes unobserved. The goal is to identify an $\varepsilon$-optimal arm with probability at least $1 - \delta$.

**Definitions.** Order the click probabilities as $w(1) \geq w(2) \geq \cdots \geq w(L)$. Assume $w(K) > w(K+1)$, so that the top $K$ items are the unique optimal set.

Define the gap for each item $i \in [L]$:

$$\Delta_i = \begin{cases} w(i) - w(K+1), & 1 \leq i \leq K, \\ w(K) - w(i), & K < i \leq L. \end{cases}$$

Define the adjusted gap for $\varepsilon \geq 0$:

$$\overline{\Delta}_i = \begin{cases} \Delta_i + \varepsilon, & 1 \leq i \leq K, \\ \Delta_K - \Delta_i + \varepsilon, & K < i \leq K', \\ \Delta_i - \varepsilon, & K' < i \leq L, \end{cases}$$

where $K' := \max\{i \in [L] : w(i) \geq w(K) - \varepsilon\}$.

Our meta-algorithm, which systematically converts fixed-confidence algorithms into fixed-budget algorithms, offers a principled way to leverage existing fixed-confidence results and extend them to fixed-budget guarantees without starting from scratch, and yields the following corollary.

**Corollary I.1.** *Assume $K' < 2K - 1$. Running Algorithm FC2FB with the base algorithm* CASCADEBAI$(\varepsilon, \cdot, K)$, *base failure rate $\delta_0 = 1/e$, $Q = 1$ and the total budget $B \geq \left(A \ln\left(\frac{1}{\delta_0}\right) + (C+1)\right) \ln\left(2\frac{A}{Q} \ln\left(\frac{1}{\delta_0}\right) + \frac{C+1}{Q}\right)$ outputs an*

*ε-optimal arm with probabiltiy at least*

$$1 - 3\exp\left(-\frac{B}{\frac{Q}{\ln(1/\delta_0)} + 4A\ln\left(\frac{B}{Q}\right)}\right)$$

*where* $A = c_1 \sum_{k=1}^{K-K_2} \frac{v_{K-k+1}^2}{\mu_{K-k+1}^2} + c_2 \frac{1}{\mu_K} \sum_{i=1}^{L-K} \frac{1}{\overline{\Delta}_{\sigma(i)}^2} + c_3 \sum_{k=2}^{2K-K'} \frac{K-k+1}{\mu_{K-k+1}} \left(\frac{1}{\overline{\Delta}_{\sigma(L-K+k)}^2} - \frac{1}{\overline{\Delta}_{\sigma(L-K+k-1)}^2}\right)$ *and* $C = c_1 \sum_{k=1}^{K-K_2} \frac{v_{K-k+1}^2}{\mu_{K-k+1}^2} \log\left(\sum_{j=1}^{K-K_2} \frac{v_{K-j+1}^2}{\mu_{K-j+1}^2}\right)$ *for some constants* $c_1, c_2, c_3$*, and* $\mu, v, \overline{\Delta}, K_2$ *are some instance-dependent constants that are defined precisely in* Zhong et al. (2020)*.*

## J. Additional Experiments

Our experiments in the main paper showed that there exists problems where our algorithm is better than SH and SHVar. However, the gap between ours and SHVar did not seem to be sufficiently large. Thus, one may argue that SHVar may actually enjoy the same performance as ours in general and that the analysis presented in Lalitha et al. (2023) is loose. While we believe their analysis is loose, we speculate that there are instances where SHVar's sample complexity can't match ours.

In this section, we provide a supporting experiment for our speculation above. Recall that SHVar operates based on two key mechanisms. First, at each stage $\ell$, it allocates samples to arms proportionally to $N_i \propto \frac{B}{\log_2(K)} \cdot \frac{\sigma_i^2}{\sum_{j \in \mathcal{A}_\ell} \sigma_j^2}$ as given in Equation (4) of Lalitha et al. (2023), where $B$ is the total budget and $\mathcal{A}_\ell$ denotes the set of active arms at the beginning of stage $\ell$. Secondly, at each stage $\ell$, SHVar eliminates the bottom half of the active arms in $\mathcal{A}_\ell$ based on their empirical mean rankings. The key idea of this experiment is to construct instances that mislead SHVar's sampling distribution such that the best arm and several near-optimal arms (i.e., those with very small suboptimality gaps) are each sampled only once. This design makes it highly likely that the best arm will be mistakenly eliminated during the halving step in the first stage, leading to poor performance. We now formally describe the instance as follows:

$$\sigma_i^2 = \frac{1}{\frac{B}{\log_2(K)} - (K-1)}, \ \forall i \in [1, \ldots, K-1]$$
$$\sigma_K^2 = 1,$$
$$\Delta_i^2 = \sigma_i^2, \ \forall i \in [2, \ldots, K].$$

where $B$ is the budget. In this instance, all arms except the last one (arm $K$) are near-optimal, each with a small suboptimal gap of $\Delta_i^2 = \frac{1}{\frac{B}{\log_2(K)} - (K-1)}$. More importantly, the best arm and the near-optimal arms ($\forall i \in [1, \ldots, K-1]$) are sampled once:

$$N_i \propto \frac{B}{\log_2(K)} \frac{\sigma_i^2}{\sum_{j \in [K]\sigma} \sigma_j^2} = \frac{B}{\log_2(K)} \frac{\frac{1}{\frac{B}{\log_2(K)} - (K-1)}}{\frac{K-1}{\frac{B}{\log_2(K)} - (K-1)} + 1} = 1$$

The remaining budget is then allocated to the last arm

$$N_K \propto \frac{B}{\log_2(K)} \frac{\sigma_K^2}{\sum_{j \in [K]\sigma} \sigma_j^2} = \frac{B}{\log_2(K)} \frac{1}{\frac{K-1}{\frac{B}{\log_2(K)} - (K-1)} + 1} = \frac{B}{\log_2(K)} - (K-1).$$

As a result, in the first stage, the last arm–which has the largest suboptimal gap ($\Delta_K^2 = \sigma_K^2 = 1$) and receives the largest number of samples–is certainly be eliminated. The best arm–being very close to the near-optimal arms and sampled only once–is also likely to be eliminated, thereby achieving our intended objective.

For comparison with our FC2FB(PE-KHN), recall that the PE-KHN algorithm (Algorithm 5) within our FC2FB framework allocates samples to arms according to $N_i = \lceil \frac{2\sigma_i^2}{\varepsilon_\ell^2} \ln\left(\frac{K}{\delta_\ell}\right) \rceil$ and eliminates arms based on the gap $\varepsilon_\ell$ ($\mathcal{S}_{\ell+1} \leftarrow \mathcal{S}_\ell \setminus \{i \in S_\ell \mid \hat{\mu}_i^{(\ell)} \leq \max_{j \in S_\ell} \hat{\mu}_j^{(\ell)} - 2\varepsilon_\ell\}$). Although the sample allocation strategies of PE-KHN and SHVar are similar, the key difference lies in their elimination rules. PE-KHN adaptively removes arms based on the gap sizes, whereas SHVar follows a fixed elimination schedule–halving the set of arms according to empirical rankings. Consequently, at the end of the first stage, half of the arms being eliminated by SHVar likely includes the best arm. In contrast, at the end of the first stage,

PE-KHN tends to eliminate only the last arm (arm $K$), and gradually removes the near-optimal arms over subsequent stages.

Furthermore, for our FC2FB(PE-KHN) algorithm, the sample complexity of this instance is

$$O\left(\frac{\sigma_1^2}{\Delta_2^2} + \sum_{i=2}^{K} \frac{\sigma_i^2}{\Delta_i^2}\right) = O\left(K\right) ,$$

which indicates that this is a relatively easy instance for our FC2FB(PE-KHN). The poor empirical performance of SHVar, illustrated in Figures 4 and 5, further reinforces the greater performance of our algorithm in comparison. Figure 4 shows the probability of misidentifying the best arm among $K = 32$ arms as the budget increases across different methods. As anticipated, SHVar's misidentification probability appears roughly the same because the best arm tends to be eliminated in the first stage, independent of the available budget. In contrast, our FC2FB(PE-KHN) method consistently identifies the best arm, achieving a misidentification probability of zero. Furthermore, the standard SH algorithm also avoids this distributional pitfall and reliably identifies the best arm, underscoring that the failure is specific to SHVar's heterogeneous sampling rule rather than to SH itself. Figure 5 shows similar trends as the number of arms $K$ increases: while SHVar's misidentification probability remains high, both FC2FB(PE-KHN) and SH maintain strong performance and successfully identify the best arm even in larger problem instances.

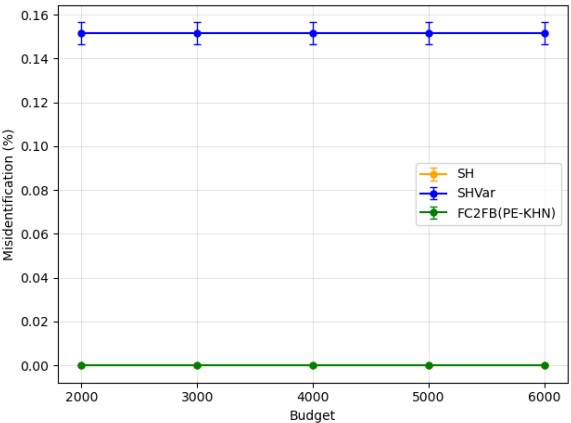

*Figure 4.* Probability of misidentifying the best arm in the Gaussian bandit as a function of the budget $B$, with $K = 32$ arms. Results are averaged over 5,000 trials.

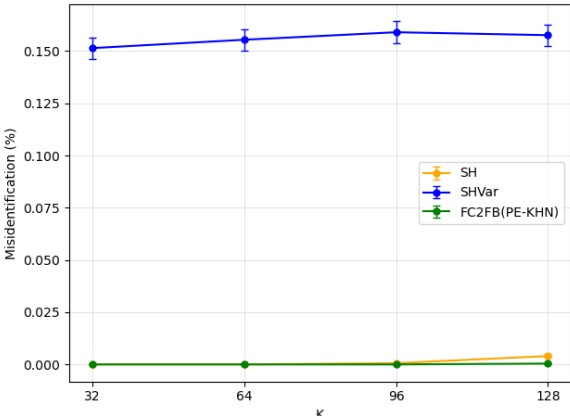

*Figure 5.* Probability of misidentifying the best arm in the Gaussian bandit as a function of the number of arms. The budget is fixed at 6000. Results are averaged over 5,000 trials.

## K. Lemmata

**Lemma K.1.** *For the budget $B$ and the number of stages $R$ defined in Algorithm 3, $A, C$ defined in Definition 3.1, $r^*$ defined in Definition B.1, $\delta_0 < 1$, and any $Q > 0$, the following statements hold.*

$$B \geq 2\left(A\ln\left(\frac{1}{\delta_0}\right) + (C+1)\right)\ln\left(2\frac{A}{Q}\ln\left(\frac{1}{\delta_0}\right) + \frac{2(C+1)}{Q}\right) \implies r^* \leq R \text{ and } B \geq 2R \cdot (C+1)$$

*Proof.*

$$r* > R \implies \max\left\{\left(R - \max\{r \in \mathbb{Z} : B' \geq T^*_{\delta_0^{2^r}}\}\right), 1\right\} > R$$

$$\implies R - \max\{r \in \mathbb{Z} : B' \geq T^*_{\delta_0^{2^r}}\} > R \qquad\qquad (R \geq 1)$$

$$\implies \max\{r \in \mathbb{Z} : B' \geq T^*_{\delta_0^{2^r}}\} < 0$$

$$\implies B' < T^*_{\delta_0} \qquad\qquad (T^*_{\delta_0} \text{ is a decreasing function in } \delta)$$

$$\implies B' < A\ln\left(\frac{1}{\delta_0}\right) + C \qquad\qquad (\text{Definition } 3.1)$$

$$\implies \lfloor\frac{B}{R}\rfloor < A\ln\left(\frac{1}{\delta_0}\right) + C$$

$$\implies \frac{B}{R} < A\ln\left(\frac{1}{\delta_0}\right) + C + 1$$

$$\implies B < R \cdot A\ln\left(\frac{1}{\delta_0}\right) + R \cdot (C+1) \qquad\qquad (\text{step (a)})$$

$$\implies B < \ln(\frac{B}{Q}) \cdot A\ln\left(\frac{1}{\delta_0}\right) + \ln(\frac{B}{Q}) \cdot (C+1)$$

$$\implies \frac{B}{Q} < \ln(\frac{B}{Q})\frac{A}{Q}\ln\left(\frac{1}{\delta_0}\right) + \ln(\frac{B}{Q})\frac{C+1}{Q}$$

$$\implies X < \ln(X)Z \qquad\qquad (\text{Let } Z := \frac{A}{Q}\ln\left(\frac{1}{\delta_0}\right) + \frac{C+1}{Q} \text{ and } X := \frac{B}{Q})$$

$$\implies X < Z\ln\left(\frac{X}{2Z} \cdot 2Z\right) = Z\ln\left(\frac{X}{2Z}\right) + Z\ln(2Z) \leq Z(\frac{X}{2Z} - 1) + Z\ln 2Z \qquad (ln(x) \leq x - 1)$$

$$\implies \frac{X}{2} < Z\ln 2Z - Z \leq Z\ln 2Z$$

$$\implies B < 2Q\left(\frac{A}{Q}\ln\left(\frac{1}{\delta_0}\right) + \frac{C+1}{Q}\right)\ln\left(2\frac{A}{Q}\ln\left(\frac{1}{\delta_0}\right) + \frac{2(C+1)}{Q}\right)$$

$$\implies B < 2\left(A\ln\left(\frac{1}{\delta_0}\right) + (C+1)\right)\ln\left(2\frac{A}{Q}\ln\left(\frac{1}{\delta_0}\right) + \frac{2(C+1)}{Q}\right)$$

Hence, by contra-position,

$$B \geq 2\left(A\ln\left(\frac{1}{\delta_0}\right) + (C+1)\right)\ln\left(2\frac{A}{Q}\ln\left(\frac{1}{\delta_0}\right) + \frac{2(C+1)}{Q}\right) \implies r^* \leq R$$

Also note that, from step (a),

$$B \geq 2\left(A\ln\left(\frac{1}{\delta_0}\right) + (C+1)\right)\ln\left(2\frac{A}{Q}\ln\left(\frac{1}{\delta_0}\right) + \frac{2(C+1)}{Q}\right)$$

$$\implies B \geq R \cdot A\ln\left(\frac{1}{\delta_0}\right) + R \cdot (C+1)$$

$$\implies B \geq 2R \cdot (C+1) \qquad\qquad (\text{Generally, } A\ln\left(\frac{1}{\delta_0}\right) \text{ is orderwise larger than } (C+1))$$

This completes the proof. □

**Lemma K.2.** *Let $\mathcal{E}$ be an event from a random trial such that $\mathbb{P}(\mathcal{E}) \leq \delta$. Let $\alpha$ satisfy $\delta < \alpha < 1$. Let $N$ be the number of trials where $\mathcal{E}$ holds true out of $L$ independent trials. Then,*

$$\mathbb{P}(N \geq \alpha L) \leq \exp\left(-\alpha L \ln\left(\frac{\alpha}{e\delta}\right)\right)$$

*Proof.* Recall the standard KL divergence based concentration inequality where $\hat{\mu}_n$ is the sample mean of $n$ Bernoulli i.i.d. random variables with head probability $\mu$:

$$\forall \ \varepsilon \geq 0, \mathbb{P}(\hat{\mu}_n - \mu \leq \varepsilon) \leq \exp(-n\mathbf{KL}(\mu + \varepsilon, \mu)) .$$

Note that $N/L$ can be viewed as the sample mean of Bernoulli trials with $\mu := \mathbb{P}(\mathcal{E})$. Then,

$$\begin{aligned}
\mathbb{P}(N \geq \alpha L) &= \mathbb{P}(\frac{N}{L} \geq \alpha) \\
&= \mathbb{P}(\frac{N}{L} - \mu \geq \alpha - \mu) \\
&\leq \exp(-L\mathbf{KL}(\alpha, \mu)) && (\alpha > \delta \geq \mu) \\
&= \exp\left(-L\left(\alpha \ln(\frac{\alpha}{\mu}) + (1-\alpha)\ln\frac{1-\alpha}{1-\mu}\right)\right) \\
&\stackrel{(a)}{\leq} \exp\left(-L\left(\alpha \ln(\frac{\alpha}{\mu}) - \alpha\right)\right) \\
&= \exp\left(-L\alpha \ln(\frac{\alpha}{e\mu})\right)
\end{aligned}$$

where $(a)$ is by the following derivation:

$$\begin{aligned}
(1-\alpha)\ln\frac{1-\alpha}{1-\mu} &= -(1-\alpha)\ln\frac{1-\mu}{1-\alpha} \\
&= -(1-\alpha)\ln\left(1 + \frac{\alpha - \mu}{1-\alpha}\right) \\
&\geq -(1-\alpha)\cdot\frac{\alpha - \mu}{1-\alpha} && (\forall \ x, \ln(1+x) \leq x) \\
&= -(\alpha - \mu) \\
&\geq -\alpha
\end{aligned}$$

Hence,

$$\mathbb{P}(N \geq \alpha L) \leq \exp\left(-L\alpha \ln(\frac{\alpha}{e\mu})\right) \leq \exp\left(-L\alpha \ln(\frac{\alpha}{e\delta})\right) && (\mu = \mathbb{P}(\mathcal{E}) \leq \delta)$$

□

