# OpenReview forum: "Fixed Budget is No Harder Than Fixed Confidence in Best-Arm Identification up to Logarithmic Factors"
_ICML.cc/2026/Conference — ICML 2026 regular_

### Official Review · Reviewer_mJFh · 2026-02-16

**Soundness:** 2
**Presentation:** 3
**Significance:** 2
**Originality:** 2
**Overall Recommendation:** 3
**Confidence:** 3

**Summary:**

This paper studies the relationship between the fixed-budget (FB) and fixed-confidence (FC) formulations of best-arm identification (BAI). The authors investigate whether one formulation is fundamentally harder than the other, with a particular focus on structured bandit settings.

The main result shows that FB is no harder than FC up to logarithmic factors. To establish this result, the authors propose a meta-algorithm, **FC2FB**, which converts any FC algorithm into an FB algorithm while preserving sample complexity guarantees. In addition, the paper introduces **FCW2S**, a procedure that converts weak FC algorithms into strong ones. The proposed framework is supported by theoretical analysis and instantiated across several structured BAI problems, accompanied by empirical results demonstrating improvements over existing FB baselines.

**Compliance With Llm Reviewing Policy:**

Affirmed.

**Final Justification:**

The clarification on the availability of strong/weak FC algorithms is helpful, but it does not change my main concern that the framework still relies on fairly restrictive non-asymptotic guarantees and does not apply to settings with only asymptotic FC results. More importantly, while the reduction is technically sound, I remain unconvinced that it delivers sufficient novelty beyond transferring guarantees from FC to FB via a combination of existing ideas.

**Key Questions For Authors:**

1. The proposed framework relies heavily on the availability of **strong FC algorithms** and **weak FC algorithms**. Could the authors clarify to what extent such algorithms are known to exist across different BAI settings, particularly in structured bandit problems? Providing concrete examples or discussing potential limitations would help assess the applicability of the framework.
2. How significant are the practical overheads introduced by FC2FB compared with directly designed FB algorithms, both theoretically (in terms of constants) and empirically?
3. How sensitive is the algorithmic performance to the choice of parameters such as $Q$ and $\delta_0$? Are there principled or adaptive strategies for tuning these parameters?
4. Can the proposed reduction framework be extended to other pure exploration settings, such as top-$k$ identification or thresholding bandits?

**Limitations:**

yes

**Strengths And Weaknesses:**

**Strengths**

- The theoretical development is well motivated and appears technically sound.
- The paper is generally well structured, with a clear problem formulation and logical progression of results.
- The work provides a unifying theoretical perspective connecting FB and FC formulations, which may facilitate transferring algorithmic advances between the two settings.

**Weaknesses**

- The practical overhead introduced by the staged reduction procedures is not fully quantified or empirically analyzed.
- The real-world impact of the framework may depend on hidden constant factors and logarithmic overheads.
- Some components of the framework build upon existing ideas such as confidence amplification and staged scheduling, which somewhat limits the level of conceptual novelty.

---

> ### Author Rebuttal · Authors · 2026-03-31
>
> * **On the availability of strong FC / weak FC algorithms:** We agree this point should be clarified better in the paper, and we will revise accordingly.
>
>     Any algorithm with non-asymptotic **high probability** sample complexity of the form $A \ln (1/\delta) + C$ is a strong FC, and the logarithmic dependence on $1/\delta$ is the dominating form of non-asymptotic high probability sample complexity. In our paper, we instantiate strong FC algorithms with concrete examples, including: heterogeneous-noise bandits via Algorithm 5 / Theorem 5.1, linear bandits, and cascading bandits. These case studies show that strong FC inputs are available in several important structured settings, and FC2FB then yields improved FB guarantees from them. In the final version, we will add more examples.
>
>     For weak FC algorithms, Section 4 explicitly broadens the applicability of the framework. Definition 4.1 allows algorithms whose guarantees are available only at a fixed confidence level. We note that any **in-expectation** sample complexity guarantee implies a weak FC guarantee simply via Markov's inequality. FCW2S then converts any such weak FC algorithm into a strong one. This is precisely why unimodal bandits are included as an application: the base guarantee there is not initially in the strong-FC form needed by FC2FB, so FCW2S is used as an intermediate step.
>
>     That being said, we admit that our current framework is not yet able to provably convert FC algorithms that only admit asymptotic sample complexity bound. We will add a discussion in detail about the availability of Strong and Weak FC algorithms and the limitation of our approach in the final version of the paper.
>
> * **On practical overheads versus directly designed FB algorithms:**
>   Please see our answer to Reviewer DEay, "Additional factors lead to a bad practical performance?".
>
> * **On the sensitivity to parameters:**
>     We agree this deserves clearer explanation.
>
>     $\delta_0$ is intended to be a fixed numerical constant such as $1/e$. To our knowledge, its effect is quite minimal as long as we are not choosing it to be too small. Its role is to define the base confidence level for the staged schedule; it is not an instance-sensitive parameter that must be finely tuned. Likewise, $Q$ serves as a lower-scale parameter ensuring that each stage has enough budget to be meaningful. The paper already recommends choosing $Q$ as the minimal scale at which the base FC algorithm begins to enjoy meaningful guarantees; for example, $Q=K$ for standard $K$-armed bandits and $Q=d$ for linear bandits, while $Q=1$ is the generic choice used in several corollaries.
>
>     Adapting $Q$ would be great idea and it could potentially lead to a better sample complexity -- this is an exciting future research direction!
>
> * **On extension beyond best-arm identification:**
>     We believe the reviewer’s suggestion is very interesting, and in fact closely aligned with the paper’s outlook.
>
>     The FC2FB meta-algorithm and its analysis only require a base algorithm satisfying the strong FC guarantee in our assumptions, and hence should carry over essentially unchanged to other pure-exploration settings such as top-$k$ identification or thresholding bandits whenever such a strong FC algorithm is available, and the success is declared as satisfying a condition or not (i.e., binary).
>
>     By contrast, FCW2S exploits the fact that in best-arm identification, each weak-FC run outputs a single arm, so aggregation can be done via a simple majority vote. For richer output spaces, where each run returns a subset, designing an aggregation rule that preserves correctness with comparable guarantees is substantially more subtle. Thus, our view is that extension to broader settings is plausible at the FC2FB level, but not automatic at the FCW2S level, and developing weak-to-strong conversions for such settings is an interesting open problem.
>
> **Final remark**
>
> The first two weaknesses mentioned by the reviewer is on empirical performance. We like to say that our main contribution is theory. To our knowledge, the first fixed budget BAI guarantee is by Audibert et al. (2010), "Best Arm ..." and the first fixed confidence BAI guarantee is by Even-dar et al. (2002), "PAC Bounds..". For more than 15 years, no one was able to relate the optimal sample complexities in these two closely-related problems. We made a great progress on this, and we did it via  (constructive) reduction of FB to FC. Previously, people often published best-arm identification papers just by proposing an FB counterpart of the FC setting, even with a worse guarantee compared to that from FC (e.g., Yang & Tan (2022), Lalitha et al. (2023)). Our reduction led to improving FB results for a number of problem settings.
>
> We hope our rebuttal help the reviewer reconsider the current score. Please let us know if you have more questions. We are happy to address.

---

> > ### Author Rebuttal · Reviewer_mJFh · 2026-04-02
> >
> > I have no further major concerns. However, I am still somewhat uncertain whether the significance of this work is sufficient for presentation at this conference. I will use the discussion phase to further assess its significance and then consider whether to raise my score.

---

> > > ### Author Response · Authors · 2026-04-06
> > >
> > > **The significance of our work:**
> > >
> > > First, our reduction is not only interesting on its own for explaining the relationship between FB and FC, but also improves bounds on a number of FB problems—keep in mind that people used to publish a paper by improving bounds on one FB problem (e.g., Yang & Tan (2022)) or two closely-related FB problems (e.g., Lalitha et al. (2023)). Our paper will provide a baseline algorithm and sample complexity upper bound for any FB identification problem for which there exists an algorithm for its FC counterpart (note: compared to FB, FC algorithms are typically easier to design and prove performance). Thus, the community will benefit a lot from knowing our result! We hope this would be reflected in your final score.
> > >
> > > Secondly, for $K$-armed bandits with a unique best arm, researchers have found that FB and FC hardness is the same up to logarithmic factors. Thus, it is natural to ask if FB and FC are equally hard up to log factors for generic identification problems. We have narrowed down the search space toward answering this question by providing one direction of the answer: FB is no harder than FC. It remains to show whether or not FC is no harder than FB. If this is true, then it would mean that FB and FC are equally hard.
> > >
> > > Hence, we believe that our reduction, which is simple and elegant, is a significant contribution in the field that compares FB and FC settings.
> > >
> > > **For better clarification on Strong and Weak FC:**
> > >
> > > We have presented a modified draft of Section 4 for Reviewer HHvH. Please be patient and refer to the revised **Section 4** in our reply to Reviewer HHvH for a better perspective on this question.

---

### Official Review · Reviewer_DEay · 2026-02-24

**Soundness:** 3
**Presentation:** 3
**Significance:** 3
**Originality:** 3
**Overall Recommendation:** 4
**Confidence:** 3

**Summary:**

This paper investigates the relationship between the fixed-confidence (FC) and fixed-budget (FB) formulations of the best-arm identification (BAI) problem in multi-armed bandits. It establishes that the fixed-budget setting is not fundamentally more difficult than the fixed-confidence setting.

The primary contribution is a result showing that FB is no harder than FC up to logarithmic factors. To achieve this, the authors propose a meta-algorithm, FC2FB, which transforms any strong fixed-confidence algorithm into a fixed-budget algorithm while preserving its high-probability sample complexity guarantee up to polylogarithmic terms (Sec. 3; Algorithm 3; Theorem 3.2). To relax the requirement of strong fixed-confidence guarantees, the paper further introduces FCW2S, a procedure that converts weak fixed-confidence algorithms into strong ones through parallel execution and majority voting (Sec. 4; Algorithm 4; Propositions 4.2–4.3). The combination of FCW2S and FC2FB provides a general framework that applies to a broad class of settings. Empirical evaluations on Gaussian bandits show that the proposed approach achieves lower misidentification probabilities (Sec. 6; Figures 1–2).

Overall, the paper establishes a general theoretical connection between the FC and FB settings and offers a practical framework for deriving fixed-budget algorithms from fixed-confidence methods.

**Compliance With Llm Reviewing Policy:**

Affirmed.

**Final Justification:**

After reading the paper and the authors’ response, I think the paper has clear strengths in originality and technical interest, especially for the fixed-budget bandit problem. I am positive about this paper overall. The work offers a novel and technically interesting idea. The paper is also generally well written, and the empirical/theoretical results support the central claims reasonably well.

Overall, I maintain my score and give a final recommendation of accept.

**Key Questions For Authors:**

1: What is the computational cost (runtime or memory) compared to the base fixed-confidence algorithm, especially for FCW2S?

2: Are there cases where the additional factors lead to a bad practical performance?

**Limitations:**

yes

**Strengths And Weaknesses:**

- Soundness

    - The framework is clearly written, and the core component is explicitly stated and analyzed (Theorem 3.2), providing explicit bounds for FC2FB (Sec. 3). The analysis carefully shows how the FC algorithm translates into FB error bounds. The weak-to-strong conversion is also rigorously justified in probabilistic guarantees (Propositions 4.2–4.3). The empirical evaluation is conducted over multiple trials and compares against standard baselines (SH, SHVar) (Sec. 6; Figures 1–2).

- Presentation

    - The paper is well written and well structured: problem formulation (Sec. 2), core reduction (Sec. 3), generalization (Sec. 4), applications (Sec. 5), and experiments (Sec. 6). Algorithms are presented explicitly, and definitions of strong and weak FC are clearly stated (Definitions 3.1 and 4.1).

- Significance

    - The result establishes a key relationship between two FC and FB. The translation framework is general and applicable across multiple settings. The ability to construct FB algorithms from FC ones could help future algorithm design and theoretical analysis.

- Originality

    - The main novelty lies in a general translation framework rather than a new bandit algorithm. The combination of FC2FB and weak-to-strong conversion provides a unified linking from FC to FB (Secs. 3–4).

---

> ### Author Rebuttal · Authors · 2026-03-31
>
> * **Computational/space complexity:**
>     Since we are taking in a blackbox algorithm, the discussion of computational/space overhead requires a computational model for the algorithms. Let us consider the model where each algorithm has a fixed initialization compute cost at the beginning (denoted by $C_{init}$), a fixed per time step cost (denoted by $C_{iter}$), and a fixed overall space complexity (denoted by space $S$). Note that there is an obvious increase in overall compute complexity if the sample complexity increases -- this is natural and we will exclude this effect from the discussion because such difference exists for any pair of algorithms with different sample complexities and can be inferred from the difference in the sample complexities.
>
>   **FC2FB** does not increase $S$ or $C_{iter}$ because it runs one instance of algorithm at a time. However, $C_{init}$ becomes multiplied by $R = \lfloor \log_2(B/Q) \rfloor$, which is moderate in our opinion.
>
>    For **FCW2S**, since we are running multiple instances in parallel, both the space complexity $S$ and $C_{init}$ increase by a factor of $L = \Theta(\ln(1/\delta))$. However, $C_{iter}$ remains the same. Thus, overall, the increase in space/time complexity is only logarithmic, which is typically considered to be mild.
>
> * **"Additional factors lead to a bad practical performance?":**
>     Our interpretation of the reviewer's question is to ask if additional factors lead to bad practical performance compared to existing FB algorithms. First of all, there are BAI problems for which there are no existing FB algorithm at all in which case we cannot say the additional factors are bad, because there is no one to compare against. So, we will consider the case where there are FB algorithms designed to directly solve the FB problem.
>
>     So, yes, in some regimes they can. It really depends on the problem-dependent complexity term. Assume that our algorithm has the leading sample complexity of $H \ln(H)$ and the state-of-the-art FB algorithm has that of $H'$. Note both $H$ and $H'$ are problem-dependent. First, if $H \approx H'$ for every problem instance (e.g., $K$-armed bandits), then our logarithmic factor blowup due to restarting multiple algorithms does influence the practical performance. Otherwise, it really depends on the problem instance. For instances where $H \ll H'$ (which can happen for linear bandits and the heterogeneous noise case, both discussed in our paper), the logarithmic factor does not incur huge cost. On the other hand, for instances where $H \approx H'$, the additional logarithmic factor does affect the practical performance. We remark that we observed that this theoretical comparison is consistent with empirical phenomenon -- the logarithmic blowup is real, but it does not become a bottleneck when the sample complexity term itself is much smaller than the FB algorithm being compared.
>
>     That said, our reduction is designed to establish a general theoretical relationship between FC and FB, not to be the final word on the most implementation-efficient FB method in every regime. It provides a generic and provably correct black-box conversion from FC to FB, showing that the FB setting is not fundamentally harder than FC up to logarithmic factors. Further, our reduction does improve the sample complexity upon existing ones in various problem settings. This is a solid theoretical finding, we believe.

---

> > ### Author Rebuttal · Reviewer_DEay · 2026-04-01
> >
> > Thanks for the detailed response. My concerns have been fully resolved, and I will maintain my original positive score.

---

> > > ### Author Response · Authors · 2026-04-06
> > >
> > > Thank you so much for your positive response! We like to reiterate the significance of our work. Our reduction is not only interesting on its own for explaining the relationship between FB and FC, but also improves bounds on a number of FB problems — keep in mind that people used to publish a paper by improving bounds on one FB problem (e.g., Yang & Tan (2022)) or two closely-related FB problems (e.g., Lalitha et al. (2023)).
> > >
> > > Our paper will provide a baseline algorithm and sample complexity upperbound for any FB identification problem for which there exists an algorithm for its FC counterpart (note: compared to FB, FC algorithms are typically easier to design and prove its performance). Thus, the community will benefit a lot from knowing our result! We hope this would be reflected in your final score.

---

### Official Review · Reviewer_SjbC · 2026-03-09

**Soundness:** 2
**Presentation:** 3
**Significance:** 3
**Originality:** 4
**Overall Recommendation:** 4
**Confidence:** 3

**Summary:**

This paper studies the best arm identification problem (BAI). Typically, this literature is divided into two parts: Fixed Confidence (FC) and Fixed Budget (FB). The fixed confidence BAI problem has been very well studied, and even asymptotically optimal algorithms have been well established, while fixed budget remains a hard and somewhat open problem.

The main contribution of the paper is to show that any reasonably good fixed confidence algorithm (Strong FC algorithm) can be translated into a fixed budget algorithm. The authors show that FC sample complexity is an upper bound of the FB sample complexity up to logarithmic factors, leading to the conclusion that FB is no harder than FC up to logarithmic factors. Furthermore, combining this meta-algorithm with existing state-of-the-art FC algorithms leads to improved sample complexity for a number of FB problems.

**Compliance With Llm Reviewing Policy:**

Affirmed.

**Final Justification:**

My questions have been adequately addressed and I am maintaining my score.

**Key Questions For Authors:**

I have addressed my concerns in the strengths and weaknesses section. To reiterate, it would be helpful if the authors could clarify whether the Strong FC algorithm is general enough to include the algorithm of Garivier and Kaufmann (2016). Additionally, the fixed budget problem appears to be hard and somewhat open, see [1] and followup of this paper. In light of this, I would kindly ask the authors to clarify what is meant by "For K-armed bandits with the unique best arm, the optimal sample complexities for both settings have been settled and they match up to logarithmic factors," along with the corresponding reference, as I may not be aware of it.

> [1] Qin, Chao. "Open Problem: Optimal Best Arm Identification with Fixed-Budget." *Conference on Learning Theory*. PMLR, 2022.

**Limitations:**

Yes

**Strengths And Weaknesses:**

## Strengths and Weaknesses

### Soundness
The submission is technically sound and the claims are well-supported by the theoretical results. That said, one concern is worth noting: the title of the paper is written in absolute generality and would ideally encompass classical results as well. In my opinion, the most celebrated fixed confidence result in the classical setting of the k-armed stochastic bandit with parametric distribution assumptions is given by Garivier and Kaufmann (2016). Their Track-and-Stop algorithm, combined with a generalized likelihood ratio test, has been shown to be asymptotically optimal with matching constants. It would be helpful if the authors could clarify whether the paper's restriction to Strong FC algorithms includes this algorithm, as this does not appear to be discussed in the paper.

### Presentation
The paper is written clearly and is well-structured. However, there is limited discussion of recent developments in the fixed budget setting and, to some extent, the fixed confidence setting for the classical stochastic *k*-armed bandit. For example, see [1].

> [1] Qin, Chao. "Open Problem: Optimal Best Arm Identification with Fixed-Budget." *Conference on Learning Theory*. PMLR, 2022.

### Significance
The paper addresses a relevant problem, meaningfully advances understanding in machine learning, and has solid potential to influence future research or practice.

### Originality
The paper presents an original idea for constructing a FB algorithm from an FC algorithm.

---

> ### Author Rebuttal · Authors · 2026-03-31
>
> We thank the reviewer for the valuable feedback and inclination towards the acceptance of our paper. We would like to address the questions and comments of the reviewer.
>
> **Garivier and Kaufmann (2016) $\cdots$:**
>
> Generally, an algorithm is expected to have a non-asymptotic guarantee to be classified as a Strong or Weak FC. An algorithm with a non-asymptotic high probability stopping time guarantee qualifies as a Strong FC. Any algorithm with a non-asymptotic expected sample complexity bound can be converted to Weak FC using Markov's inequality (Corollary 5.5). Any Weak FC can be converted to Strong FC using FCW2S.
>
> The original Track-and-Stop (Garivier & Kaufmann, 2016) algorithm provides only an asymptotic bound. Hence, this cannot be classified as either a strong or a weak FC algorithm, because both classifications need a non-asymptotic bound. However, Poiani et al. (2025) develop an extension called ``Sticky Track-and-Stop'', analyze it in a non-asymptotic regime, and provide a non-asymptotic expected sample complexity bound. Any algorithm with a non-asymptotic expected sample complexity bound will be a weak FC at least. (This can be achieved by applying Markov's inequality, similar to Corollary 5.5). Hence, it can be converted to Strong FC using our proposed FCW2S framework.
>
> **[1] Qin, Chao. "Open Problem: Optimal Best Arm Identification with Fixed-Budget $\cdots$:**
>
>  Thank you for suggesting this work. This work is inspired by the existence of an algorithm class with matching asymptotic upper and lower bounds for the FC setting. This work raises an open question whether it is possible to have such matching bounds for FB settings (with a very specific form for that bound, including constant factors)?
>
> This is, in fact, one of the weaknesses of our work in the sense that our framework establishes a **no harder** sample complexity in the non-asymptotic regime. In the asymptotic regime of $\delta \rightarrow 0$, the presence of $\log(\log \frac{1}{\delta})$ denies similar asymptotic guarantees, even though doubly logarithmic factor is effectively a numerical constant for the range of $\delta$ that people typically care (e.g., if $\delta = 10^{-10}$, then $\ln(\ln(1/\delta)) \approx 3.14$).
>
> So, our framework cannot answer the open problem, since the asymptotic version of our guarantee has a doubly logarithmic factor in terms of error probability on the right-hand side.
>
> **For K-armed bandits with the unique best arm, the optimal sample complexities for both settings have been $\cdots$:**
>
> This comparison between FC and FB for multi-arm bandits is detailed in Carpentier & Locatelli (2016). They have shown that FB algorithms will incur a sample complexity of $H \log (K)$ order for at least one instance of the bandit problem with an unknown problem parameter $H$. Thereby, they claim that the Successive Rejection algorithm from Audibert et al. (2010) is an optimal algorithm in fixed-budget settings. In that sense, the optimal sample complexity of the FB setting $H \log K$ and the FC setting $H$ match up to logarithmic factors.
>
> Hence, is it natural to think that the above finding from Carpentier & Locatelli (2016) solves the open problem? We guess that it does not solve the open problem because,
>
> 1. Even after Carpentier's findings, the upper bound and lower bound do not exactly match. (Audibert et al. (2010) upper bound $H_2 \log (K)$ and Carpentier's lower bound $H \log (K)$). Meanwhile, Qin's formulation expects a match **even up to constant factors**.
>
> For the fixed budget setting, the optimal constant factors for the $\ln(1/\delta)$ term are not known, but the optimality ignoring constant factors are known up to logarithmic factors. Nevertheless, it is an interesting research question.
>
> Furthermore, we will expand our related works section to add more clarifications regarding these asymptotic and non-asymptotic differences, discussing how our work relates to Qin (2022), Carpentier & Locatelli (2016), and Garivier & Kaufmann (2016) in the final version. We will add discussion on the candidates of Strong and Weak FC and how this categorization relates to the existing types of guarantees (asymptotic, high probability, expected sample complexity, etc).
>
> Please let us know if you have more questions. We are happy to address.
>
> Audibert, J.-Y., Bubeck, S., and Munos, R. Best Arm Identification in Multi-Armed Bandits.
>
> Carpentier, A. and Locatelli, A. Tight (lower) bounds for the fixed budget best arm identification bandit
> problem.
>
> Degenne, R. On the existence of a complexity in fixed budget bandit identification.
>
> Garivier, A. and Kaufmann, E. Optimal best arm identification with fixed confidence.
>
> Kaufmann, E., Capp´e, O., and Garivier, A. On the complexity of best-arm identification in multi-armed
> bandit models.
>
> Poiani, R., Bernasconi, M., and Celli, A. Non-asymptotic analysis of (sticky) track-and-stop.

---

> > ### Author Rebuttal · Reviewer_SjbC · 2026-04-04
> >
> > Thank you for the detailed response. My questions have been adequately addressed and I am maintaining my positive score.

---

> > > ### Author Response · Authors · 2026-04-06
> > >
> > > Thank you so much for your positive response! We like to reiterate the significance of our work. Our reduction is not only interesting on its own for explaining the relationship between FB and FC, but also improves bounds on a number of FB problems — keep in mind that people used to publish a paper by improving bounds on one FB problem (e.g., Yang & Tan (2022)) or two closely-related FB problems (e.g., Lalitha et al. (2023)).
> > >
> > > Our paper will provide a baseline algorithm and sample complexity upperbound for any FB identification problem for which there exists an algorithm for its FC counterpart (note: compared to FB, FC algorithms are typically easier to design and prove its performance). Thus, the community will benefit a lot from knowing our result! We hope this would be reflected in your final score.

---

### Official Review · Reviewer_HHvH · 2026-03-16

**Soundness:** 3
**Presentation:** 1
**Significance:** 2
**Originality:** 3
**Overall Recommendation:** 3
**Confidence:** 3

**Summary:**

The paper studies a fundamental question in best-arm identification: how are fixed-budget and fixed-confidence settings related? The authors propose a novel algorithm FC2FB which takes in FC algorithm as a black-box and turns it into an FB algorithm, and shows this enjoys a sample complexity which matches upto logarithmic factors of sample complexity of the FC algorithm.

**Compliance With Llm Reviewing Policy:**

Affirmed.

**Key Questions For Authors:**

1. What about is FC harder than FB? The paper introduction talks about exploring the relation, but focuses exclusively on one direction.
2. Also, is the result Th. 3.2 tight?

**Limitations:**

Yes

**Strengths And Weaknesses:**

**Strengths**
1. The proposed algorithm and analysis goes beyond simply running FC and truncate, by removing dependence on unknown instance hardness parameters intrinsic to FC
2. Sec 1-3  are clear and succinct

**Weaknesses**
1. The experiments fail to capture insights proposed in the paper, and contains few inconsistencies. For example, in text, it says K=64 and in Fig. 1, it says K=32.
2. Several hyperparameters introduced in the paper are unclear, such as Q, \delta_0 etc.
3. Alg 4 is not clear at all and should be updated. Some clarity is in the appendix, but certain details are still missing (like what are votes, I can only guess). Can you explain the updated algorithm in detail?
4. The writing feels very disorganized from Sec 4 of the paper.

---

> ### Author Rebuttal · Authors · 2026-03-31
>
> We thank the reviewer for the feedback. We would like to clarify some of the concerns and questions.
>
> **The experiments fail $\cdots$:**  It is a typo -- it should be $K=32$; we'll fix it. We did not understand why our experiments fail to capture insight. Do you mind providing more details so we can address it? Our experiments empirically prove that indeed our reduction results in better FB algorithms compared to the existing state-of-the-art algorithm (e.g., Figure 1,2,4,5).
>
> **Several hyperparameters introduced $\cdots$:**
>
> $\delta_0$ is the base failure rate. That is the failure rate of the black box FC. One can easily set it to be like $.5$ or $1/e$ with no other disadvantage, but we wanted to keep it generic. Regarding $Q$, we did try to explain it carefully right below Algorithm 3. Please let us know in which aspect this was not clear, and we will do our best to explain it. Basically, one can set it to be $1$, but there are better recommendations for specific problems. Still, it would only improve logarithmic factors (and the main scope of our work is not to be concerned about the logarithmic factors anyway, as often done in learning theory papers).
>
> **Alg 4 is not clear at all  $\cdots$:**
> 1.  Algorithm 4 basically runs L independent instances of a weak FC with $\delta_0$ base failure rate in parallel. (Line 223)
>
> 2. Here, parallel means each instance gets to pull the arm and observe the reward in the round-robin method. This is taken care of by the internal (clock) time step. (Line 226).
>
> 3. If any of these instances self-terminates (stopping condition of the FC met), we take note of the output arm $\hat{J}_{\ell}$ (this is one vote for that arm) and eliminate the instance from the round-robin. (stop running that instance).  (Line 230-233).
>
> 4. If half of all the initiated instances self-terminate, then we stop the algorithm and start to count votes. (Line 234 - Line 237).
>
> Here, the votes $v_i$ mean the number of instances that output arm $i$. Whoever wins this voting is the final output ($\hat{J}$).
> From an election point of view, we have $L$ polling centers (instances), and if half of them have finished voting, then we count the winners (arms) in each of the completed polling centers. Whoever wins in the most polling centers is the final winner ($\hat{J}$).
>
> Furthermore, you can relate this algorithm to the so-called **boosting** or **amplification**, where weak learners are used to create a strong guarantee. Here, we also use weak FC to create a strong guarantee. When we use multiple instances, the probability of getting wrong output from more than half of the instances decays exponentially. This results in an exponential improvement required for Strong FC.
>
> We will add these in the final version.
>
> **The writing feels very disorganized $\cdots$**:
>
> We will improve the organization. Further, we will move our application of weak to strong conversion from Section 5.4 to Section 4.
>
> **What about is FC harder than $\cdots$:**
>
> First of all, if we ignore logarithmic factors (which is the scope we set up for our paper), FB is no harder than FC, so FC cannot be harder than FB.
>
> Thus, our understanding is that the reviewer is asking how much FC is harder than FB if we include logarithmic factors. To answer this, we can consider the special case of unstructured bandits (i.e., multi-armed bandits). In this setting, Carpentier & Locatelli (2016) proved that there exists an instance for which the FB has to suffer an additional factor of $\log K$ compared to the optimal FC sample complexity.
>
> But again, our main scope is ignoring logarithmic factors, as is done frequently in learning theory work.
>
> **Also, is the result Th. 3.2 tight? $\cdots$:**
>
> We believe the result is tight up to logarithmic factors, because we think it would be impossible to have even smaller sample complexity guarantee compared to that of the base FC algorithm being used. So, the question is if the logarithmic factors are tight.
>
> While it is possible that our logarithmic factor of log(A/Q) might be improvable (either improved analysis or improved algorithm), we claim that some logarithmic factors have to be there for the following reasons:
>
> 1. Without any logarithmic factor, our bound would go against the lower bound for the K-armed bandit problem (i.e., log(K) factor proved in Carpentier & Locatelli (2016)).
> 2. Having an additional log factor is the only way that our result can co-exist with the impossibility results of Degenne (2023).
>
> Hope our answers clarified the reviewer's concerns. We hope that the reviewer will raise their score. Please let us know if you have more questions.
>
> Carpentier, A. and Locatelli, A. Tight (lower) bounds for the fixed budget best arm identification bandit
> problem.
>
> Degenne, R. On the existence of a complexity in fixed budget bandit identification.
>
> Qin, C. Open problem: Optimal best arm identification with fixed budget

---

> > ### Author Rebuttal · Reviewer_HHvH · 2026-04-04
> >
> > Thank you for answering my questions. For the experiments, I was expecting to see how FC2FB compare for different blackbox FC algorithms. There is no comparison for FC2FB(VD-BESTARMID) as well. My questions about the tightness of the logarithmic factors has been questioned satisfactorily. I now have a better understanding of Alg. 4 though I stand with my previous comment that the writing feels very disorganized after Sec. 4. A rewrite would definitely have me increase the score.

---

> > > ### Author Response · Authors · 2026-04-06
> > >
> > > **Correction: what about FC harder than FB:**
> > >
> > > Please ignore our previous answer; there were some inaccuracies. The reviewer was probably asking ``what about is FC **no** harder than FB..'' Note that it might be true that FC and FB are equally hard, ignoring logarithmic factors. The scope of our work is to show that FB is no harder than FC (ignoring log factors), so investigating whether or not FC (ignoring log factors) is no harder than FB would be an interesting future research direction!
> > >
> > > **A rewrite would definitely have me increase the $\cdots$:**
> > >
> > > Sorry for brevity because of space constraint.
> > >
> > > Organization from sec 3 onward.
> > >
> > > 1. (Sec 3) Our paper constructs a framework to convert FC algorithms to an FB algorithm with some nice guarantees. $FC2FB(\text{Strong FC}) \rightarrow \text{A good FB}$
> > >
> > > 2. However, the framework requires a strong FC as input. (Def 3.1). The majority of FC algorithms with **Non-asymptotic high probability bounds** fall into this category.
> > >
> > > 3. (Sec 4) We do not want to limit our framework's scope. So we come up with an algorithm FCW2S ($FCW2S(\text{Weak FC}) \rightarrow \text{Strong FC}$). **Weak FC covers all the FC algorithms with non-asymptotic guarantees, including those with only an expected sample complexity guarantee.**
> > >
> > > 4. (Sec 5) We show examples, such that $FC2FB(\text{Strong FC/ Weak FC })$, has better performance than the best existing FB algorithms.
> > >
> > > **Section 4:**
> > >
> > > **Swift move from sec 3 to sec 4 before sec 5:**
> > >
> > > In the previous section, we presented our framework that transforms any Strong FC algorithms to an FB algorithm with the same sample complexity, but with just a logarithmic overhead. This conversion is very useful in domains where the existing state-of-the-art FB algorithms have a considerably worse sample complexity compared to the best FC algorithms available. Before moving on to discuss those domains where our framework is useful (Section 5), we will first discuss the limitations of our FC2FB framework and how we can overcome them.
> > >
> > > **Scope \& limitations of FC2FB:**
> > >
> > > As of now, our FC2FB framework requires a Strong FC as an input. Any fixed confidence algorithm with non-asymptotic high probability sample complexity bound with the **logarithmic dependence** on $\frac{1}{\delta}$ is a Strong FC. A majority of the fixed confidence algorithms fall into the Strong FC category. Just to name a few, [will provide a few references]. However, we do not want to limit our framework to just Strong FCs. What about FC algorithms with only **expected sample complexity** bound or sample complexity bounds with **polynomial dependence on $\frac{1}{\delta}$**? Let us call those algorithms Weak FC.
> > >
> > > **Introduction of Weak FC:**
> > >
> > > In this section, we present another framework that transforms an algorithm with any non-asymptotic sample complexity bound (Weak FC) to a strong FC. Let us first define Weak FC as follows.
> > >
> > > Please see Def 4.1.
> > >
> > > So, by definition, Weak FC covers FC algorithms with sample complexity bounds with **polynomial dependence on $\frac{1}{\delta}$**. The main merit of the Weak FC definition is that any FC algorithm with an **expected sample complexity bound** can be converted to weak FC by applying Markov's inequality. That is,
> > >
> > > [We will add Markov's inequality for expected to high probability as in Corollary 5.5]
> > >
> > > **Scope of Weak FC:**
> > >
> > > [refs to algs with expected sample complexity from the literature].
> > >
> > > **Algorithm and guarantees for Weak FC $\rightarrow$ Strong FC:**
> > >
> > > Now, we present the FCW2S algorithm that transforms any weak FC into a Strong FC.
> > >
> > > Please see Alg 4 and Prop 4.2 and 4.3.
> > >
> > > $$P(\tau > \frac{4 \cdot f (\delta_0)}{\ln 1/(4e\delta_0)} \cdot \ln \frac{1}{\delta}) < \delta. $$ (we will add this for more clarity)
> > >
> > > **Discussion on expanded scope and remaining limitations of FC2FB:**
> > >
> > > Hence, FCW2S facilitates us to apply the FC2FB framework for any FC that has a non-asymptotic sample complexity bound. That being said, our framework FC2FB cannot be applied to FC algorithms that admit only an asymptotic sample complexity bound (Garivier \& Kaufmann, 2016). [mention future direction].
> > >
> > > **Section 5:**
> > >
> > > In this sec, we analyze some bandit domains where our framework FC2FB will have a better bound than the best available FB algorithms in the respective domains. This is possible because, in these domains, the existing FC algorithms have superior bounds compared to the existing FB algorithms.
> > >
> > > **Sec 5.1 - 5.3**:  **FC2FB**(**Strong FC**} $\in$ $\{\text{PE-KHN, VD-BESTARMID, FC-Peace}\}$) $\implies$ **FB** with better guarantee than **existing standard FB** $\in \{\text{SH, SHVar, SHAdaVar, VBR, OD-LinBAI}\}$. (for heterogeneous and Linear bandits)
> > >
> > > **Sec 5.4**:  **FC2FB**(FCW2S(**Weak FC** $\in \{ \text{UniTT} \}$)) $\implies$ **FB** with better guarantee than **existing standard FB** $\in \{\text{FB-BAUB }\}$. (for Unimodal bandits)
> > >
> > > Welcome the reviewer's feedback on organization and will include experiments in the final version.

---

### Decision · Program_Chairs · 2026-04-30

**Decision:**

Accept (regular)

**Comment:**

This work proposed a novel algorithm FC2FB which takes in FC algorithm as a black-box and turns it into an FB algorithm, and shows this enjoys a sample complexity which matches upto logarithmic factors of sample complexity of the FC algorithm. As FH is still a field where the best arm identification has not been fully resolved, the findings of this work is quite interesting. Meanwhile, author(s) are suggested to further improve the manuscript by incorporating reviews.